# Boosting GWAS using biological networks: A study on susceptibility to familial breast cancer

Héctor Climente-González[1,2,3,4]*, Christine Lonjou[1,2,3], Fabienne Lesueur[1,2,3], GENESIS study group[¶], Dominique Stoppa-Lyonnet[5,6,7‡], Nadine Andrieu[1,2,3], Chloé-Agathe Azencott[1,2,3]

**1** Institut Curie, PSL Research University, Paris, France, **2** INSERM, U900, Paris, France, **3** MINES ParisTech, PSL Research University, CBIO-Centre for Computational Biology, Paris, France, **4** RIKEN Center for Advanced Intelligence Project (AIP), Tokyo, Japan, **5** Service de Génétique, Institut Curie, Paris, France, **6** INSERM, U830, Paris, France, **7** Université Paris Descartes, Paris, France

‡For the GENESIS study group
¶ Membership of the GENESIS study group is listed in the Acknowledgments.
* hector.climente@riken.jp

**Data Availability Statement:** We cannot share genotype data publicly for confidentiality reasons, but are available from GENESIS. Interested

## Abstract

Genome-wide association studies (GWAS) explore the genetic causes of complex diseases. However, classical approaches ignore the biological context of the genetic variants and genes under study. To address this shortcoming, one can use biological networks, which model functional relationships, to search for functionally related susceptibility loci. Many such network methods exist, each arising from different mathematical frameworks, pre-processing steps, and assumptions about the network properties of the susceptibility mechanism. Unsurprisingly, this results in disparate solutions. To explore how to exploit these heterogeneous approaches, we selected six network methods and applied them to GENESIS, a nationwide French study on familial breast cancer. First, we verified that network methods recovered more interpretable results than a standard GWAS. We addressed the heterogeneity of their solutions by studying their overlap, computing what we called the *consensus*. The key gene in this consensus solution was *COPS5*, a gene related to multiple cancer hallmarks. Another issue we observed was that network methods were unstable, selecting very different genes on different subsamples of GENESIS. Therefore, we proposed a *stable consensus* solution formed by the 68 genes most consistently selected across multiple subsamples. This solution was also enriched in genes known to be associated with breast cancer susceptibility (*BLM*, *CASP8*, *CASP10*, *DNAJC1*, *FGFR2*, *MRPS30*, and *SLC4A7*, P-value = $3 \times 10^{-4}$). The most connected gene was *CUL3*, a regulator of several genes linked to cancer progression. Lastly, we evaluated the biases of each method and the impact of their parameters on the outcome. In general, network methods preferred highly connected genes, even after random rewirings that stripped the connections of any biological meaning. In conclusion, we present the advantages of network-guided GWAS, characterize their shortcomings, and provide strategies to address them. To compute the

researchers can contact Séverine Eon-Marchais (severine.eon-marchais@curie.fr).

**Funding:** C.-A.A. received funding from Agence Nationale de la Recherche (ANR-18-CE45-0021-01). H.C.-G. was supported from the European Union's Horizon 2020 research and innovation program (Marie Skłodowska-Curie [666003]). Financial support for GENESIS resource and genotyping was provided by the Ligue Nationale contre le Cancer (grants PRE05/DSL and PRE07/DSL, to D.S.-L.; and grant PRE11/NA, to N.A.), the French National Institute of Cancer (grant INCa grant b2008-029/LL-LC, to D.S.-L.) and the comprehensive cancer center SiRIC (Site de Recherche Intégrée sur le Cancer: grant INCa-DGOS-4654, to N.A.) The funders had no role in study design, data collection and analysis, decision to publish, or preparation of the manuscript.

**Competing interests:** The authors have declared that no competing interests exist.

consensus networks, implementations of all six methods are available at https://github.com/hclimente/gwas-tools.

## Author summary

Genome-wide association studies (GWAS) scan thousands of genomes to identify variants associated with a complex trait. Over the last 15 years, GWAS have advanced our understanding of the genetics of complex diseases, and in particular of cancers. However, they have led to an apparent paradox: the more we perform such studies, the more it seems that the entire genome is involved in every disease. The omnigenic model offers an appealing explanation: only a limited number of *core* genes are directly involved in the disease, but gene functions are deeply interrelated, and so many other genes can alter the function of the core genes. These interrelations are often modeled as networks, and multiple algorithms have been proposed to use these networks to identify the subset of core genes involved in a specific trait. This study applies and compares six such network methods on GENESIS, a GWAS dataset for familial breast cancer in the French population. Combining these approaches allows us to identify potentially novel breast cancer susceptibility genes and provides a mechanistic explanation for their role in the development of the disease. We provide ready-to-use implementations of all the examined methods.

## 1 Introduction

In human health, genome-wide association studies (GWAS) aim at quantifying how single-nucleotide polymorphisms (SNPs) predispose to complex diseases, like diabetes or some forms of cancer [1]. To that end, in a typical GWAS, thousands of unrelated samples are genotyped: the cases, suffering from the disease of interest, and the controls, taken from the general population. Then, a statistical test of association (e.g., based on logistic regression) is conducted between each SNP and the phenotype. Those SNPs with a P-value lower than a conservative Bonferroni threshold are candidates to further studies in independent cohorts. Once the risk SNPs have been discovered, they can be used for risk assessment and deepening our understanding of the disease.

GWAS have successfully identified thousands of variants underlying many common diseases [2]. However, this experimental setting also presents inherent challenges. Some of them stem from the high dimensionality of the problem, as every GWAS to date studies more variants than samples are genotyped. This limits the statistical power of the experiment, as it can only detect variants with larger effects [3]. This is particularly problematic since the prevailing view is that most genetic architectures involve many variants with small effects [3]. Additionally, to avoid false positives, most GWAS apply a conservative multiple test correction, typically the previously mentioned Bonferroni correction. However, Bonferroni correction is overly conservative when the statistical tests correlate, as happens in GWAS [4]. Another open issue is the interpretation of the results, as the functional consequences of most common variants are unknown. On top of that, recent large-sampled studies suggest that numerous loci spread all along the genome contribute to a degree to any complex trait, in accordance with the infinitesimal model [5]. The recently proposed omnigenic model [6] offers an explanation: genes are strongly interrelated and influence each other's function, which allows alterations in most genes to impact the subset of "core" genes directly involved in the disease's mechanism.

Hence, a comprehensive statistical framework that includes the structure of biological data might help alleviate the issues above.

For this reason, many authors turn to network biology to handle the complex interplay of biomolecules that lead to disease [7, 8]. As its name suggests, network biology models biology as a network, where the biomolecules under study, often genes, are nodes, and selected functional relationships are edges that link them. These relationships come from evidence that the genes jointly contribute to a biological function; for instance, their expressions are correlated, or their products establish a protein-protein interaction. Under this view, complex diseases are not the consequence of a single altered gene, but of the interaction of multiple interdependent molecules [9]. In fact, an examination of biological networks shows that disease genes have differential properties [9, 10]: they tend to occupy central positions in the network (although not the most central ones); disease genes for the same pathology tend to cluster in modules; and often they are bottlenecks that interconnect modules.

Network-based discovery methods exploit the differential properties described above to identify disease genes using GWAS data [11, 12]. In essence, each gene receives a score of association with the disease, computed from the GWAS data, and a set of biological relationships, given by a network built on prior knowledge. Then, the problem becomes finding a functionally-related set of highly-scoring genes. Multiple solutions have been proposed to this problem, often stemming from different mathematical frameworks and considerations of what the optimal solution looks like. For example, some methods restrict the problem to specific types of subnetworks. Such is the case of LEAN [13], which focuses on "star" subnetworks, i.e., instances where both a gene and its direct interactors are associated with the disease. Other algorithms, like dmGWAS [14] and heinz [15], do not impose such strong constraints and search for subnetworks interconnecting genes with high association scores. However, they differ in their tolerance to the inclusion of low-scoring nodes and the topology of the solution. Lastly, other methods also consider the topology of the network, favoring groups of nodes that are not only high-scoring but also densely interconnected; such is the case of HotNet2 [16], SConES [17], and SigMod [18].

In this work, we studied the relevance of network-based approaches to genetics by applying these six network methods to GWAS data. They use different interpretations of the omnigenic model and provide a representative view of the field. We worked on the GENESIS dataset [19], a study on familial breast cancer conducted in the French population. After a classical GWAS approach, we used these network methods to identify additional breast cancer susceptibility genes. Lastly, we compared the solutions obtained by the different methods and studied their intersection to obtain consensus solutions of predisposition to familial breast cancer that addressed their shortcomings.

## 2 Materials and methods

### 2.1 GENESIS dataset, preprocessing, and quality control

The GENE Sisters (GENESIS) study investigated risk factors for familial breast cancer in the French population [19]. Index cases were patients with infiltrating mammary or ductal adenocarcinoma, who had a sister with breast cancer, and tested negative for *BRCA1* and *BRCA2* pathogenic variants. Controls were unaffected colleagues or friends of the cases born around the year of birth of their corresponding case (± 3 years). We focused on the 2 577 samples of European ancestry, of which 1 279 were controls, and 1 298 were cases. The genotyping platform was the iCOGS array, a custom Illumina array designed to study the genetic susceptibility to hormone-related cancers [20]. It contained 211 155 SNPs, including SNPs putatively associated with breast, ovarian, and prostate cancers, SNPs associated with survival after diagnosis,

and SNPs associated to other cancer-related traits, as well as candidate functional variants in selected genes and pathways.

We discarded SNPs with a minor allele frequency lower than 0.1%, those not in Hardy–Weinberg equilibrium in controls (P-value < 0.001), and those with genotyping data missing on more than 10% of the samples. We also removed a subset of 20 duplicated SNPs in *FGFR2*. We excluded the samples with more than 10% missing genotypes. After controlling for relatedness, we excluded 17 additional samples (6 for sample identity error, 6 controls related to other samples, 2 cases related to an index case, and 3 additional controls having a high relatedness score). Lastly, based on study selection criteria, 11 other samples were removed (1 control having cancer, 4 index cases with no affected sister, 3 half-sisters, 1 sister with lobular carcinoma *in situ*, 1 with a *BRCA1* or *BRCA2* pathogenic variant detected in the family, 1 with unknown molecular diagnosis). The final dataset included 1 271 controls, 1 280 cases, and 197 083 SNPs.

We looked for population structure that could produce spurious associations. A principal component analysis revealed no visual differential population structure between cases and controls (S1 Fig). Independently, we did not find evidence of genomic inflation ($\lambda = 1.05$) either, further confirming the absence of confounding population structure.

## 2.2 SNP- and gene-based GWAS

To measure the association between genotype and susceptibility to breast cancer, we performed a per-SNP 1 d.f. $\chi^2$ allelic test using PLINK v1.90 [21]. To obtain significant SNPs, we performed a Bonferroni correction to keep the family-wise error rate below 5%. The threshold used was $\frac{0.05}{197083} = 2.54 \times 10^{-7}$. The summary statistics of the analyzed SNPs are available in S1 Table.

Then, we used VEGAS2 [22] to compute the gene-level association score from the P-values of the SNPs mapped to them. More specifically, we mapped SNPs to genes through their genomic coordinates: all SNPs located within the boundaries of a gene, ±50 kb, were mapped to that gene. We computed VEGAS2 scores for each gene using only the 10% of SNPs with the lowest P-values among all those mapped to it. We used the 62 193 genes described in GENCODE 31 [23], although only 54 612 mapped to at least one SNP. Out of those, we focused exclusively on the 32 767 that had a gene symbol. Out of the 197 083 SNPs remaining after quality control, 164 037 mapped to at least one of these genes. We also performed a Bonferroni correction to obtain significant genes; in this case, the threshold of significance was $\frac{0.05}{32767} = 1.53 \times 10^{-6}$. The summary statistics of the analyzed genes are available in S2 Table.

## 2.3 Network methods

**2.3.1 Mathematical notations.**   In this article, we used undirected, vertex-weighted networks, or graphs, $G = (V, E, w)$. $V = \{v_1, \ldots, v_n\}$ refers to the vertices (or nodes), with weights $w : V \to \mathbb{R}$. Equivalently, $E \subseteq \{\{x, y\} | x, y \in V \land x \neq y\}$ refers to the edges. When referring to a subnetwork $S$, $V_S$ is the set of nodes in $S$ and $E_S$ is the set of edges in $S$. A special case of subgraphs are *connected* subgraphs, which occur when every node in the subgraph can be reached from any other node.

Nodes can be described by properties provided by the topology of the graph. We focused on the betweenness centrality, or *betweenness*: the number of times a node participates in a shortest path between two other nodes. We normalized the betweenness by dividing it by $\frac{(N-1)(N-2)}{2}$, where $N$ is the number of nodes.

We also used two matrices that describe two different properties of a graph. Both matrices are square and have as many rows and columns as nodes are in the network. The element $(i, j)$ hence represents a relationship between $v_i$ and $v_j$. The *adjacency matrix* $W_G$ contains a 1 when the corresponding nodes are connected, and 0 otherwise; its diagonal is zero. The *degree matrix* $D_G$ is a diagonal matrix that contains the number of edges of each node.

**2.3.2 Networks.   Gene networks**: The mathematical formulations of the different network methods are compatible with any type of biological network (e.g., from protein interactions or gene co-expression). Here, we used protein-protein interaction networks (PPIN) for all gene-centric network methods, as PPINs are interpretable, well-characterized, and the methods were designed to run efficiently on them. We built our PPIN from both binary and co-complex interactions stored in the HINT database (release April 2019) [24]. Unless otherwise specified, we used only interactions coming from high-throughput experiments, leaving out targeted studies that might bias the topology of the PPIN. Out of the 146 722 interactions from high-throughput experiments that HINT stores, we could map 142 541 to a pair of gene symbols, involving 13 619 genes. 12 880 of those mapped to a genotyped SNP after quality control, involving 127 604 interactions. The scoring function for the nodes changed from method to method (Section 2.3.3).

Additionally, we compared the results obtained on this PPIN with those obtained on a PPIN built using interactions coming from both high-throughput and targeted studies. In that case, out of the 179 332 interactions in HINT, 173 797 mapped to a pair of gene symbols. Out of those, 13 735 mapped to a genotyped SNP after quality control, involving 156 190 interactions.

**SNP networks**: SConES [17] was the only network method designed to handle SNP networks. As in gene networks, two SNPs were connected in a SNP network when there was evidence of shared functionality between them. Azencott et al. [17] proposed three ways of building such networks: connecting the SNPs consecutive in the genomic sequence ("GS network"); interconnecting all the SNPs mapped to the same gene, on top of GS ("GM network"); and interconnecting all SNPs mapped to two genes for which a protein-protein interaction exists, on top of GM ("GI network"). We focused on the GI network using the PPIN described above, as it fitted the scope of this work better. However, at different stages, we also compared GI to GS and GM to understand how including the PPIN affects SConES' output. For the GM network, we used the mapping described in Section 2.3.5. In all three, we scored the nodes using the 1 d.f. $\chi^2$ statistic of association. The properties of these three subnetworks are available in S3 Table.

**2.3.3 High-score subnetwork search algorithms.**   Genes that contribute to the same function are nearby in the PPIN and can be topologically related to each other in diverse ways (densely interconnected modules, nodes around a hub, a path, etc.). Several aspects have to be considered when developing a network method: how to score the nodes, whether the affected mechanisms form a single connected component or several, how to frame the problem in a computationally efficient fashion, which network to use, etc. Unsurprisingly, multiple solutions have been proposed. We examined six of them: five that explore the PPIN, and one which explores SNP networks. We selected open-source methods that had an implementation available and accessible documentation. We summarize their main differences in Table 1. We scored both SNPs and genes with the P-values (or transformations) computed in Section 2.2.

**dmGWAS** dmGWAS seeks the subgraph with the highest local density in low P-values [14].

To that end, it searches candidate solutions using a greedy, "seed and extend", heuristic:

1. Select a seed node *i* and form the subnetwork $S_i = \{i\}$.

**Table 1. Summary of the differences between the network methods.**

| Method | Field | Nodes | Exhaustive | Solution | Comp. | Input | Scoring | Ref. |
|---|---|---|---|---|---|---|---|---|
| dmGWAS | GWAS | Genes | No | - | 1 | Summary | $-\log_{10}(P)$ | [14] |
| heinz | Omics | Genes | Yes | - | 1 | Summary | BUM | [15] |
| HotNet2 | Omics | Genes | Yes | Module | $\geq 1$ | Summary | Local FDR | [16] |
| LEAN | Omics | Genes | Yes | Star | $\geq 1$ | Summary | $-\log_{10}(P)$ | [13] |
| SConES | GWAS | SNPs | Yes | Module | $\geq 1$ | Genotypes | 1 d.f. $\chi^2$ | [17] |
| SigMod | GWAS | Genes | Yes | Module | $\geq 1$ | Summary | $\Phi^{-1}(1 - P)$ | [18] |

*Field*: field in which the algorithm was developed. *Nodes*: the type of nodes in the network, either genes (PPIN) or SNPs. *Exhaustive*: whether the method explores all the possible solutions given the selected parameters. *Solution*: additional properties enforced on the solution, other than containing high scoring, connected nodes. *Comp.*: number of connected components in the solution. *Input*: genotype data or GWAS summary statistics. *Scoring*: how SNP/gene P-values were transformed into node scores. In the case of heinz, BUM stands for beta-uniform model; for SigMod, $\Phi^{-1}$ represents the inverse of the cumulative distribution function of the standard Normal distribution. *Ref.*: original publication featuring the algorithm.

2. Compute Stouffer's Z-score $Z_m$ for $S_i$ as

$$Z_m = \frac{1}{\sqrt{k}}\sum_{j \in S_i} z_j, \tag{1}$$

where $k$ is the number of genes in $S_i$, $z_j$ is the Z score of gene $j$, computed as $\phi^{-1}(1 - \text{P-value}_j)$, and $\phi^{-1}$ is the inverse normal distribution function.

3. Identify neighboring nodes of $S_i$, i.e., nodes at distance $\leq$ `d`.

4. Add the neighboring nodes whose inclusion increases $Z_{m+1}$ by more than a threshold $Z_m \times (1 + $ `r`$)$.

5. Repeat 2-4 until no further enlargement is possible.

6. Add $S_i$ to the list of subnetworks to return. Normalize its Z-score as

$$Z_N = \frac{Z_m - \text{mean}(Z_m(\pi))}{\text{SD}(Z_m(\pi))}, \tag{2}$$

where $Z_m(\pi)$ represents a vector containing 100 000 random subsets of the same number of genes.

DmGWAS carries out this process on every gene in the PPIN. We used the implementation of dmGWAS in the dmGWAS 3.0 R package [25]. Unless otherwise specified, we used the suggested parameters `d` = 2 and `r` = 0.1. We used the function `simpleChoose` to select the solution, which aggregates the top 1% subnetworks.

**heinz** The goal of heinz is to identify the highest-scored connected subnetwork [15]. The authors proposed a transformation of the genes' P-value into a score that is negative under weak association with the phenotype, and positive under a strong one. This transformation is achieved by modeling the distribution of P-values by a beta-uniform model (BUM) parameterized by the desired false discovery rate (FDR). Thus formulated, the problem is NP-complete, and hence solving it would require a prohibitively long computational time. To solve it efficiently, it is re-cast as the Prize-Collecting Steiner Tree Problem, which seeks to select the connected subnetwork S that maximizes the *profit* p(S), defined as:

$$p(S) = \sum_{v \in V_S} p(v) - \sum_{e \in E_S} c(e). \tag{3}$$

were $p(v) = w(v) - w'$ is the *profit* of adding a node, $c(e) = w'$ is the *cost* of adding an edge, and $w' = min_{v \in V_G} w(v)$ is the smallest node weight of $G$. All three are positive quantities. Heinz implements the algorithm from Ljubić et al. [26] which, in practice, is often fast and optimal, although neither is guaranteed. We used BioNet's implementation of heinz [27, 28].

**HotNet2** HotNet2 was developed to find connected subgraphs of genes frequently mutated in cancer [16]. To that end, it considers both the local topology of the PPIN and the nodes' scores. An insulated heat diffusion process captures the former: at initialization, the score of the node determines its initial heat; iteratively each node yields heat to its "colder" neighbors and receives heat from its "hotter" neighbors while retaining part of its own (hence, *insulated*). This process continues until a stationary state is reached, in which the temperature of the nodes does not change anymore, and results in a diffusion matrix $F$. $F$ is used to compute the similarity matrix $E$ that models exchanged heat as

$$E = F \, \mathrm{diag}(w(V)), \tag{4}$$

where $\mathrm{diag}(w(V))$ is a diagonal matrix with the node scores in its diagonal. For any two nodes $i$ and $j$, $E_{ij}$ models the amount of heat that diffuses from node $j$ to node $i$. Hence, $E_{ij}$ can be interpreted as a (non-symmetric) similarity between those two nodes. To obtain densely connected solutions, HotNet2 prunes $E$, only preserving edges such that $w(E) > \delta$. Lastly, HotNet2 evaluates the statistical significance of the solutions by comparing their size to the size of PPINs obtained by permuting the node scores. We assigned the initial node scores as in Nakka et al. [29], giving a 0 to the genes unlikely to be truly associated with the disease, and $-\log_{10}(\text{P-value})$ to those likely to be. In the GENESIS dataset, the threshold separating both was a P-value of 0.125, which we obtained using a local FDR approach [30]. HotNet2 has two parameters: the restart probability $\beta$, and the threshold heat $\delta$. Both parameters are set automatically by the algorithm, which is robust to their values [16]. HotNet2 is implemented in Python [31].

**LEAN** LEAN searches altered "star" subnetworks, that is, subnetworks composed of one central node and all its interactors [13]. By imposing this restriction, LEAN can exhaustively test all such subnetworks (one per node). For a particular star subnetwork of size $m$ LEAN performs three steps:

1. Rank the P-values of the involved nodes as $p_1 \leq \ldots \leq p_m$.

2. Conduct $k$ binomial tests to compute the probability of having $k$ out of $m$ P-values lower or equal to $p_k$ under the null hypothesis. The minimum of these $k$ P-values is the score of the subnetwork.

3. Transform this score into a P-value through an empirical distribution obtained via a subsampling scheme, where gene sets of the same size are selected randomly, and their score computed.
   We adjust these P-values for multiple testing through a Benjamini-Hochberg correction. We used the implementation of LEAN from the `LEANR` R package [32].

**SConES** SConES searches the minimal, modular, and maximally associated subnetwork in a SNP graph [17]. Specifically, it solves the problem

$$\arg\max_{S \subseteq G} \underbrace{\sum_{v \in V_S} w(v)}_{\text{association}} - \lambda \underbrace{\sum_{v \in V} \sum_{u \notin V_S} W_{vu}}_{\text{connectivity}} - \underbrace{\eta |V_S|}_{\text{sparsity}}, \tag{5}$$

where λ and η are parameters that control the sparsity and the connectivity of the model. The connectivity term penalizes disconnected solutions, with many edges between selected and unselected nodes. Given a λ and an η, Eq 5 has a unique solution that SConES finds using a graph min-cut procedure. As in Azencott et al. [17], we selected λ and η by cross-validation, choosing the values that produce the most stable solution across folds. In this case, the selected parameters were η = 3.51, λ = 210.29 for SConES GS; η = 3.51, λ = 97.61 for SConES GM; and η = 3.51, λ = 45.31 for SConES GI. We used the version on SConES implemented in the R package `martini` [33, 34].

**SigMod** SigMod searches the highest-scoring, most densely connected subnetwork [18]. It addresses an optimization problem similar to that of SConES (Eq 5), but with a different connectivity term that favors solutions containing many edges:

$$\arg \max_{S \in G} \underbrace{\sum_{v \in V_S} w(v)}_{\text{association}} + \lambda \underbrace{\sum_{v \in V} \sum_{u \in V_S} W_{vu}}_{\text{connectivity}} - \underbrace{\eta |V_S|}_{\text{sparsity}}. \tag{6}$$

As for SConES, this optimization problem can also be solved by a graph min-cut approach. SigMod presents three important differences with SConES. First, it was designed for PPINs. Second, it favors solutions containing many edges between the selected nodes. SConES, instead, penalizes connections between selected and unselected nodes. Third, it explores the grid of parameters differently, and processes their respective solutions. Specifically, for the range of λ = $\lambda_{min}$, . . ., $\lambda_{max}$ for the same η, it prioritizes the solution with the largest change in size from $\lambda_n$ to $\lambda_{n+1}$. Additionally, that change needs to be larger than a user-specified threshold `maxjump`. Such a large change implies that the network is densely interconnected. This results in one candidate solution for each η, which is processed by removing any node not connected to any other. A score is assigned to each candidate solution by summing their node scores and normalizing by size. Finally, SigMod chooses the candidate solution with the highest standardized score, and that is not larger than a user-specified threshold (`nmax`). We used the default parameters `maxjump` = 10 and `nmax` = 300. Sig-Mod is implemented in an R package [35].

**Consensus** We built a consensus solution by retaining the genes selected by at least two of the six methods (using SConES GI for SConES). It includes any edge between the selected genes in the PPIN.

We performed all the computations in the cluster described in Section 2.8.

**2.3.4 Parameter space.** We used the network methods with the parameters recommended by their authors, or with the default parameters in their absence. Additionally, we explored the parameter space of the different methods to study how they alter the output.

**dmGWAS** We tested multiple values for `r` (0.0001, 0.001, 0.01, 0.05, 0.1, 0.25, 0.5, and 1) and `d` (1, 2, and 3).

**heinz** We tested multiple FDR thresholds (0.05, 0.1, 0.15, 0.2, 0.25, 0.3, 0.35, 0.4, 0.45, 0.5, 0.55, 0.6, 0.65, 0.7, 0.75, 0.8, 0.85, 0.9, 0.95, 1).

**HotNet2** We tested different thresholds to decide which genes would receive a score of 0 and which ones a score of $-log_{10}$(P-value): 0.001, 0.01, 0.05, 0.125, 0.25, and 0.5.

**LEAN** We used the following significance cutoffs for LEAN's P-values (0.05, 0.1, 0.15, 0.2, 0.25, 0.3, 0.35, 0.4, 0.45, 0.5, 0.55, 0.6, 0.65, 0.7, 0.75, 0.8, 0.85, 0.9, 0.95, and 1).

**SConES** We used the values of λ and η that `martini` explores by default (35.54, 5.40, 0.82, 0.12, 0.02, 0.01, 4.39e-4, 6.68e-5, 1.02e-5, and 1.55e-6 in both cases)

**SigMod** We tested multiple values for the parameters `nmax` (10, 50, 100, 300, 700, 1000, and 10 000) and `maxjump` (5, 10, 20, 30, and 50).

**2.3.5 Comparing SNP-based methods to gene-based methods and vice versa.** In multiple steps of this article, we compared the outcome of a method that works on genes with the outcome of one that works on SNPs. For this purpose, we used the SNP-gene correspondence described in Section 2.2. To convert a list of SNPs into a list of genes, we included all the genes mapped to any of those SNPs. Conversely, to convert a list of genes into a list of SNPs, we included all the SNPs mapped to any of those genes.

## 2.4 Pathway enrichment analysis

We searched for pathways enriched in the solution of each network method. We conducted a hypergeometric test on pathways from Reactome [36] using the function `enrichPathway` from the `ReactomePA` R package [37]. The universe of genes included any gene that we could map to a SNP in the iCOGS array (Section 2.2). We adjusted the P-values for multiple testing as in Benjamini and Hochberg [38] (BH): pathways with a BH adjusted P-value < 0.05 were deemed significant.

## 2.5 Benchmark of methods

We evaluated multiple properties of the different methods (described in Sections 2.5.1 and 2.5.2) through a 5-fold subsampling setting. We applied each method to 5 random subsets of the original GENESIS dataset containing 80% of the samples (*train set*). When pertinent, we evaluated the solution on the remaining 20% (*test set*). We used the 5 repetitions to estimate the average and the standard deviation of the different measures. Every method and repetition was run in the same computational settings (Section 2.8).

**2.5.1 Properties of the solution.** We compared the runtime, the number of selected features (genes or SNPs), and the stability (sensitivity to the choice of train set) of the different network methods. Nogueira and Brown [39] proposed quantifying a method's stability using the Pearson correlation between the genes selected on different subsamples. This correlation was calculated between vectors with the length of the total number of features, containing a 0 at position *i* if feature *i* was not selected and a 1 if it was.

**2.5.2 Classification accuracy of selected SNPs.** A desirable solution offers good predictive power on the unseen test samples. We evaluated the predicting power of the SNPs selected by the different methods through the performance of an L1-penalized logistic regression classifier, which searches for a small subset of SNPs that provides good classification accuracy at predicting the outcome (case/control). The L1 penalty helps to account for linkage disequilibrium by reducing the number of SNPs included in the model (*active set*). The active set was a plausible, more sparse solution with comparable predictive power to the original solution. The L1 penalty was set by cross-validation, choosing the value that minimized misclassification error.

We applied each network method to each train set. Then, we trained the classifier on the same train set using only the selected SNPs. When the method retrieved a list of genes, we proceeded as explained in Section 2.3.5. Lastly we evaluated the sensitivity and the specificity of the classifier on the test set. To obtain a baseline, we also trained the classifier on all the SNPs of the train set.

We did not expect a linear model on selected SNPs to separate cases from controls well. Indeed, the lifetime cumulative incidence of breast cancer among women with a family history of breast or ovarian cancer, and no *BRCA1/2* mutations, is only 3.9 times more than in the general population [40]. However, classification accuracy may be one additional informative criterion on which to evaluate solutions.

## 2.6 Comparison to state-of-the-art

An alternative way to evaluate the methods is by comparing their solutions to an external dataset. For that purpose, we used the 153 genes associated to familial breast cancer on DisGeNET [41]. Across this article, we refer to these genes as *breast cancer susceptibility genes*.

Additionally, we used the summary statistics from the Breast Cancer Association Consortium (BCAC), a meta-analysis of case-control studies conducted in multiple countries. BCAC included 13 250 641 SNPs genotyped or imputed on 228 951 women of European ancestry, mostly from the general population [42]. Through imputations, BCAC includes more SNPs than the iCOGS array used for GENESIS (Section 2.1). However, in all the comparisons in this paper we focused on the SNPs that passed quality control in GENESIS. Hence, we used the same Bonferroni threshold as in Section 2.2 to determine the significant SNPs in BCAC. We also computed gene-scores in the BCAC data using VEGAS2, as in Section 2.1. In this case, we did use the summary statistics of all 13 250 641 available SNPs and the genotypes from European samples from the 1000 Genomes Project [43] to compute the LD patterns. Since these genotypes did not include chromosome X, we excluded it from this analysis. All comparisons included only the genes in common between GENESIS and BCAC, so we used a different Bonferroni threshold ($1.66 \times 10^{-6}$) to call gene significance.

## 2.7 Network rewirings

Rewiring the PPIN while preserving the number of edges of each gene allowed to study the impact of the topology on the output of network methods. Indeed, the edges lose their biological meaning while the topology of the network is conserved. We produced 100 such rewirings by randomly swapping edges in the PPIN. We still scored the genes as described in Section 2.3.3. We applied only four methods on the rewirings: heinz, dmGWAS, LEAN, and SigMod. We excluded HotNet2 and SConES since they took notably longer to run. As on the real PPIN, LEAN did not produce significant results on any of the rewirings either.

## 2.8 Computational resources

We ran all the computations on a Slurm cluster, running Ubuntu 16.04.2 on the nodes. The CPU models on the nodes were Intel Xeon CPU E5-2450 v2 at 2.50GHz and Intel Xeon E5-2440 at 2.40GHz. The nodes running heinz and HotNet2 had 20GB of memory; the ones running dmGWAS, LEAN, SConES, and SigMod, 60GB. For the benchmark (Section 2.5), we ran each of the methods on the same Ubuntu 16.04.2 node, with a CPU Intel Xeon E5-2450 v2 at 2.50GHz, and 60GB of memory.

## 2.9 Code and data availability

We developed computational pipelines for several steps of GWAS analyses, such as physically mapping SNPs to genes, computing gene scores, and running six different network methods. We created a pipeline with a clear interface that should work on any GWAS dataset for each of those processes. They are compiled in https://github.com/hclimente/gwas-tools. The code that applies them to GENESIS, as well as the code that reproduces all the analyses in this article are

available at https://github.com/hclimente/genewa. We deposited all the produced gene solutions on NDEx (http://www.ndexbio.org), under the UUID e9b0e22a-e9b0-11e9-bb65-0ac135e8bacf.

Summary statistics for SNPs and genes are available at S1 and S2 Tables, respectively. We cannot share genotype data publicly for confidentiality reasons, but are available from GENESIS. Interested researchers can contact Séverine Eon-Marchais (severine.eon-marchais(at)curie.fr).

# 3 Results

## 3.1 Conventional SNP- and gene-based analyses retrieve the *FGFR2* locus in the GENESIS dataset

We conducted association analyses in the GENESIS dataset (Section 2.1) at both SNP and gene levels (Section 2.2). At the SNP level, two genomic regions had a P-value lower than the Bonferroni threshold on chromosomes 10 and 16 (S2A Fig). The former overlaps with the gene *FGFR2*, the latter with *CASC16* and the protein-coding gene *TOX3*. Variants in both *FGFR2* and *TOX3* have been repeatedly associated with breast cancer susceptibility in other case-control studies [42], in studies on *BRCA1* and *BRCA2* carriers [44], and in hereditary breast and ovarian cancer families negative for mutations in *BRCA1* and *BRCA2* [45]. At the gene level, only *FGFR2* was significantly associated with breast cancer (S2B Fig).

Closer examination revealed two other regions (3p24 and 8q24) having low, albeit not genome-wide significant, P-values. Both of them have been associated with breast cancer susceptibility in the past [46, 47]. We applied an L1-penalized logistic regression using all GENESIS genotypes as input and the phenotype (cancer/healthy) as the outcome (Section 2.5.2). The algorithm selected 100 SNPs, both from all regions mentioned above and new ones (S2C Fig). However, it was unclear why those SNPs were selected, as emphasized by the high P-value of some of them, which further complicates the biological interpretation. Moreover, and in opposition to what would be expected under the omnigenic model, the genes to which these SNPs map (Section 2.3.5) were not interconnected in the protein-protein interaction network (PPIN, Section 2.3.2). Moreover, the classification performance of the model was low (sensitivity = 55%, specificity = 55%, Section 2.5). Together, these issues motivated exploring network methods, which consider not only statistical association but also the location of each gene in a PPIN to find susceptibility genes.

## 3.2 Network methods successfully identify genes associated with breast cancer

We applied six network methods to the GENESIS dataset (Section 2.3.3). As none of the networks examined by LEAN was significant (Benjamini-Hochberg [BH] correction adjusted P-value < 0.05), we obtained five solutions (Fig 1): one for each of the remaining four gene-based methods, and one for SConES GI (which works at the SNP level).

These solutions differed in many aspects, making it hard to draw joint conclusions. For starters, the overlap between the genes featured in each solution was relatively small (Fig 1A). However, the methods tended to agree on the genes with the strongest signal: genes selected by more methods tended to have lower P-value of association (Fig 1B).

Another major difference was the solution size: the largest solution, produced by HotNet2, contained 440 genes, while heinz's contained only 4 genes. While SConES GI did not recover any protein coding gene, working with SNP networks rather than gene networks allowed it to

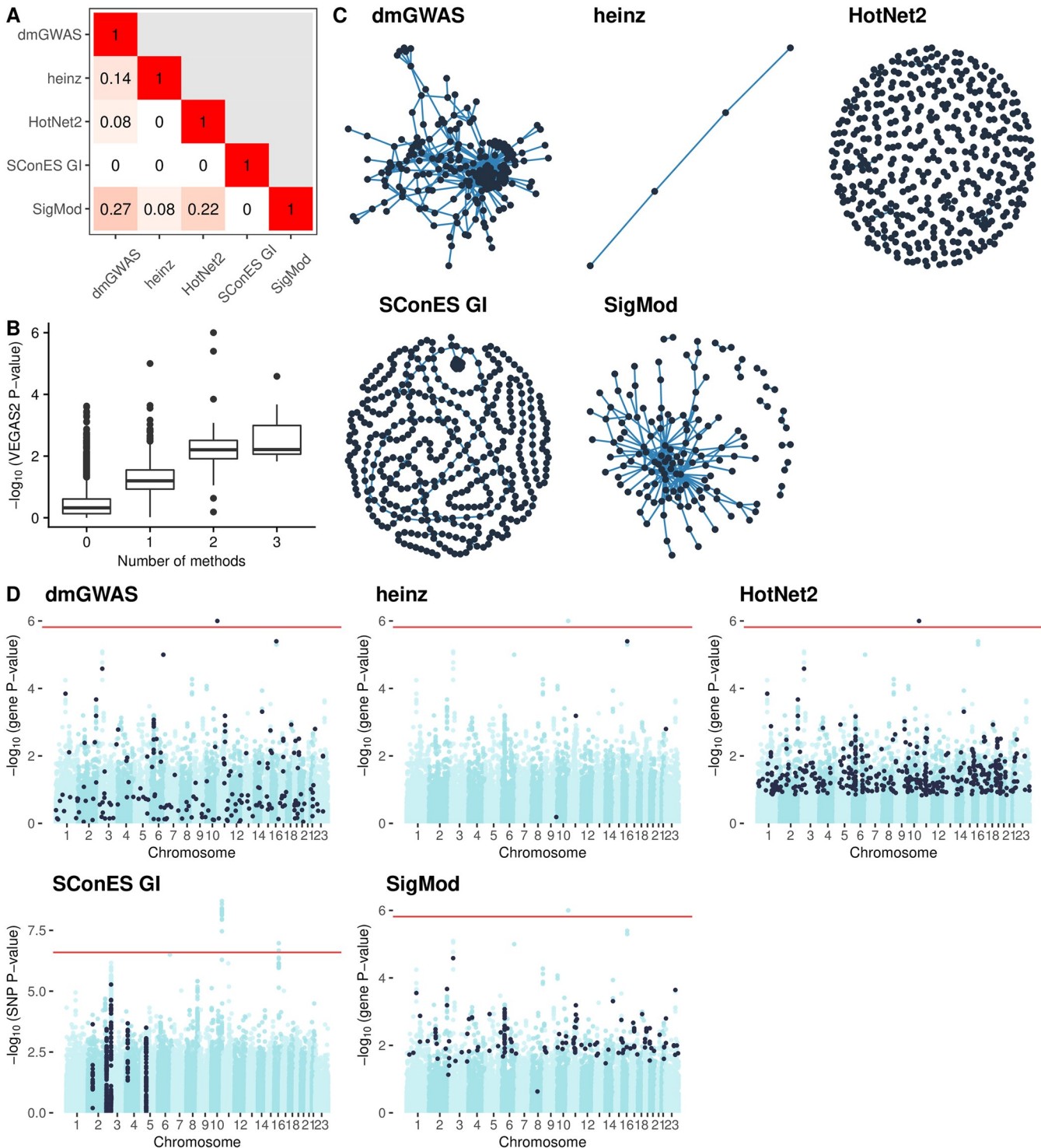

**Fig 1. Overview of the solutions produced by the different network methods (Section 2.3.3) on the GENESIS dataset.** As LEAN did not produce any significant solution (BH adjusted P-value < 0.05), it is not shown. Unless indicated otherwise, results refer to SNPs for SConES GI, and to genes for the other methods. **(A)** Overlap between the genes selected by each method, measured by Pearson correlation between indicator vectors (Sections 2.5.1 and 2.3.5). **(B)** Distribution of VEGAS2 P-values of the genes in the PPIN not selected by any network method (12 213), and of those selected by 1 (575), 2 (73), or 3 (20) methods. **(C)** Solution networks produced by the different methods. **(D)** Manhattan plots of SNPs/genes; in black, the method's solution. The red line indicates the Bonferroni threshold ($2.54 \times 10^{-7}$ for SNPs, $1.53 \times 10^{-6}$ for genes).

**Table 2. Summary statistics on the solutions of multiple network methods on the PPIN.** The first row contains the summary statistics on the whole PPIN.

| Network | # genes | # edges | # components | $\overline{\text{Betweenness}}$ | $\hat{P}_{gene}$ | # genes in consensus |
|---|---|---|---|---|---|---|
| HINT HT | 13 619 | 142 541 | 15 | $1.8 \times 10^{-4}$ | 0.46 | 93/93 |
| dmGWAS | 194 | 450 | 1 | $5.3 \times 10^{-4}$ | 0.19 | 55/93 |
| heinz | 4 | 3 | 1 | $1.2 \times 10^{-3}$ | 0.001 | 4/93 |
| HotNet2 | 440 | 374 | 130 | $8.3 \times 10^{-5}$ | 0.048 | 63/93 |
| LEAN | 0 | 0 | 0 | - | - | 0/93 |
| SConES GI | 0 (1) | 0 | 0 | - | - | 0/93 |
| SigMod | 142 | 249 | 11 | $1 \times 10^{-3}$ | 0.008 | 84/93 |
| Consensus | 93 | 186 | 21 | $5.5 \times 10^{-4}$ | 0.006 | 93/93 |
| Stable consensus | 68 | 49 | 32 | $1 \times 10^{-3}$ | 0.005 | 43/93 |

**# genes**: number of genes selected out of those that are part of the PPIN; for SConES GI, the total number of genes, including RNA genes, was added in parentheses.

**# components**: number of connected components. $\overline{\text{Betweenness}}$: median (normalized) betweenness of the selected genes in the PPIN. $\hat{P}_{gene}$: median VEGAS2 P-value of the selected genes. **# genes in consensus**: number of genes in common between the method's solution and the 93 genes in the consensus solution.

retrieve four subnetworks in intergenic regions and another subnetwork overlapping an RNA gene (*RNU6-420P*).

The topologies of the five solutions differed as well (Fig 1C), as measured by the median betweenness and the number of connected components (Table 2). Three methods yielded more than one connected component: SConES, as described above, SigMod, and HotNet2. HotNet2 produced 135 subnetworks, 115 of which have fewer than five genes. The second largest subnetwork (13 nodes) contained the two breast cancer susceptibility genes *CASP8* and *BLM* (Section 2.6).

Lastly, pathway enrichment analyses on each method's solution (Section 2.4) also revealed similarities and differences between them. It linked different parts of SigMod's solution to four processes (S4 Table): protein translation (including mitochondrial), mRNA splicing, protein misfolding, and keratinization (BH adjusted P-values < 0.03). Interestingly, the dmGWAS solution (S5 Table) was also related to protein misfolding (*attenuation phase*, BH adjusted P-value = 0.01). However, it additionally included proteins related to mitosis, DNA damage, and regulation of TP53 (BH adjusted P-values < 0.05), which match previously known mechanisms of breast cancer susceptibility [48]. As with SigMod, the genes in HotNet2's solution (S6 Table) were involved in mitochondrial translation (BH adjusted P-value = $1.87 \times 10^{-4}$), but also in glycogen metabolism and transcription of nuclear receptors (BH adjusted P-value < 0.04).

Despite their differences, there were additional common themes. All obtained solutions had lower association P-values than the whole PPIN (median VEGAS2 P-value $\ll$ 0.46, Table 2), despite containing genes with higher P-values as well (Fig 1D). This illustrates the trade-off between controlling for type I error and biological relevance. However, there are nuances between solutions in this regard: heinz strongly favored genes with lower P-values, while dmGWAS was less conservative (median VEGAS2 P-values 0.0012 and 0.19, respectively); SConES tended to select whole LD-blocks; and HotNet2 and SigMod were less likely to select low scoring genes.

Additionally, the solutions presented other desirable properties. First, four of them were enriched in known breast cancer susceptibility genes (dmGWAS, heinz, HotNet2, and SigMod, Fisher's exact test one-sided P-value < 0.03). Second, the genes in three solutions displayed, on average, a significantly higher betweenness centrality than the rest of the genes (dmGWAS, HotNet2, and SigMod, Wilcoxon rank-sum test P-value < $1.4 \times 10^{-21}$). This agrees

with the notion that disease genes are more central than other non-essential genes [49], an observation that holds in breast cancer (one-tailed Wilcoxon rank-sum test P-value = $2.64 \times 10^{-5}$ when comparing the betweenness of known susceptibility genes versus the rest). Interestingly, the SNPs in SConES' solution were also more central than the average SNP (S3 Table), suggesting that causal SNPs are also more central than non-associated SNPs.

### 3.3 A case study: The consensus solution

Despite their shared properties, the differences between the solutions suggested that each of them captured different aspects of cancer susceptibility. Indeed, out of the 668 genes that were selected by at least one method, only 93 were selected by at least two, 20 by three, and none by four or more. Encouragingly, the more methods selected a gene, the higher its association score to the phenotype (Fig 1B), a relationship that plateaued at 2. Hence, to leverage their strengths and compensate for their respective weaknesses, we built a consensus solution using the genes shared among at least two solutions (Section 2.3.3). This solution (Fig 2A) contained 93 genes and exhibited the aforementioned properties of the individual solutions: enrichment in breast cancer susceptibility genes and higher betweenness centrality than the rest of the genes.

A pathway enrichment analysis of the genes in the consensus solution also showed similar pathways as the individual solutions (S7 Table). We found two involved mechanisms: *mitochondrial translation* and *attenuation phase*. The former is supported by genes like *MRPS30* (VEGAS2 P-value = 0.001), which encode a mitochondrial ribosomal protein and was also linked to breast cancer susceptibility [50]. Interestingly, increased mitochondrial translation has been found in cancer cells [51], and its inhibition was proposed as a therapeutic target. With regards to the attenuation phase of heat shock response, it involved three Hsp70 chaperones: HSPA1A, HSPA1B, and HSPA1L. The genes encoding these proteins are all near each other at 6p21, in the region known as HLA. In fact, out of the 22 SNPs mapped to any of these three genes, 9 mapped to all three, and 4 to two, which made it hard to disentangle their effects. *HSPA1A* was the most strongly associated gene (VEGAS2 P-value = $8.37 \times 10^{-4}$).

Topologically, the consensus consisted of a connected component composed of 49 genes and multiple smaller subnetworks (Fig 2B and 2C). Among the latter, 19 genes were in subnetworks containing a single gene or two connected nodes. This implied that they did not have a consistently altered neighborhood but were strongly associated themselves and hence picked by at least two methods. The large connected component contained genes that are highly central in the PPIN. This property weakly anticorrelated with the P-value of association to the disease (Pearson correlation coefficient = -0.26, S3 Fig). This anticorrelation suggested that these genes were selected because they were on the shortest path between two high scoring genes. Because of this, we hypothesize that highly central genes might contribute to the heritability through alterations of their neighborhood, consistent with the omnigenic model of disease [6]. For instance, the most central node in the consensus solution was *COPS5*, a component of the COP9 signalosome that regulates multiple signaling pathways. *COPS5* is related to multiple hallmarks of cancer and is overexpressed in multiple tumors, including breast and ovarian cancer [52]. Despite its lack of association in GENESIS or in studies conducted by the Breast Cancer Association Consortium (BCAC) [42] (VEGAS2 P-value of 0.22 and 0.14, respectively), its neighbors in the consensus solution had consistently low P-values (median VEGAS2 P-value = 0.006).

### 3.4 Network methods boost discovery

We compared the results obtained with different network methods to the European sample of BCAC, the largest GWAS to date on breast cancer (Section 2.6). Although BCAC case-control

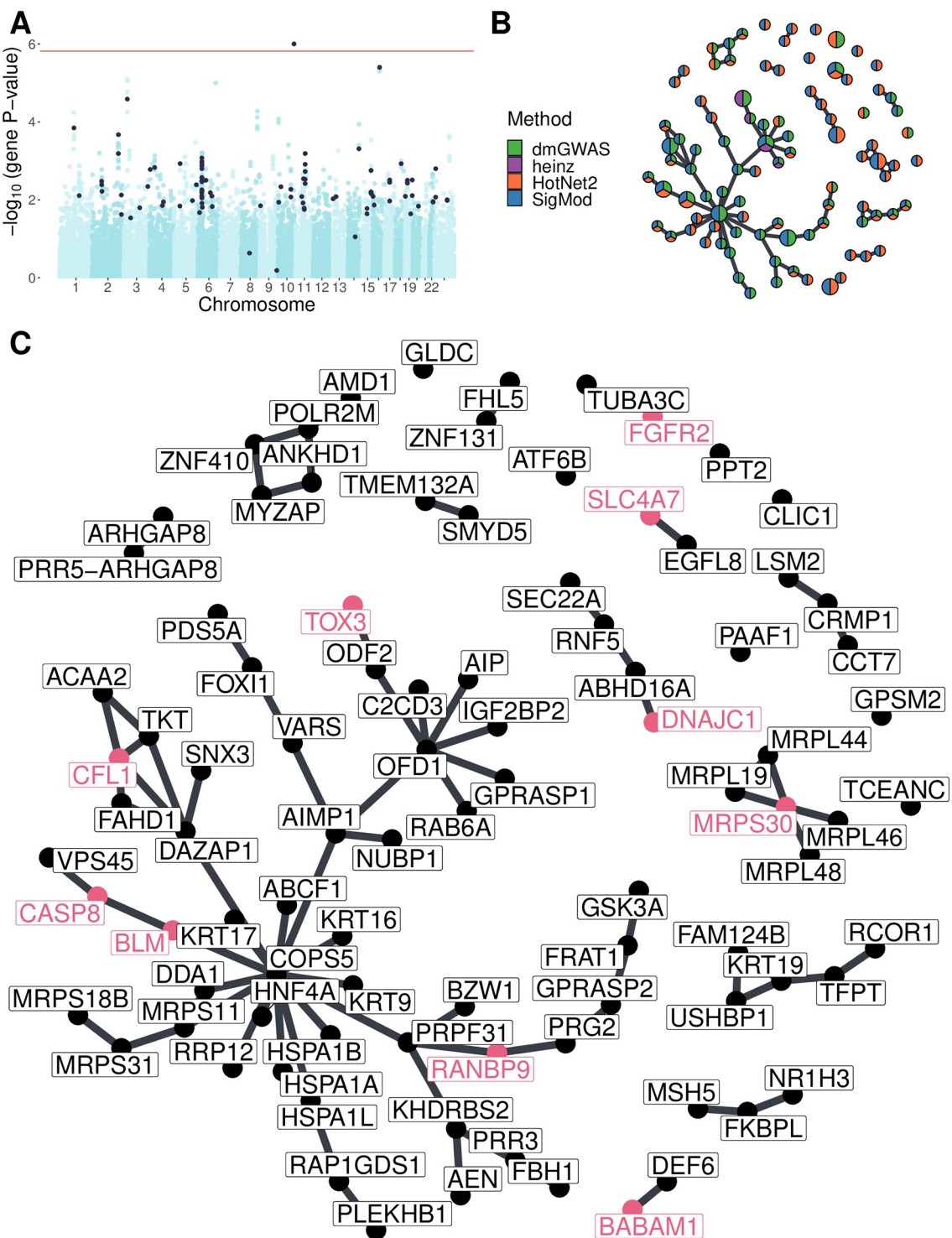

**Fig 2. Consensus solution on GENESIS (Section 2.3.3). (A)** Manhattan plot of genes; in black, the ones in the consensus solution. The red line indicates the Bonferroni threshold ($1.53 \times 10^{-6}$ for genes). **(B)** Consensus network. Each gene is represented by a pie chart, which shows the methods that selected it. We enlarged the two most central genes (*COPS5* and *OFD1*), the known breast cancer susceptibility genes, and the BCAC-significant genes (Section 2.6). **(C)** The nodes are in the same disposition as in panel B, but we indicated every gene name. We colored in pink the names of known breast cancer susceptibility genes and BCAC-significant genes.

studies do not necessarily target cases with a familial history of breast cancer like GENESIS does, this comparison is pertinent since we expect a shared genetic architecture at the gene level, at which most network methods operate. Together with BCAC's scale (90 times more samples than GENESIS), this shared genetic architecture provided a reasonable counterfactual of what we would expect if GENESIS had a larger sample size. We computed a gene association score on BCAC (Section 2.6). The solutions provided by the different network methods overlapped significantly with BCAC hits (Fisher's exact test P-value < 0.019). The gene-based methods achieved comparable precision (2%-25%) and recall (1.3-12.1%) at recovering BCAC-significant genes (S4A Fig). Interestingly, while SConES GI achieved a similar recall at the SNP-level (8.6%), it showed a much higher precision (47.3%).

### 3.5 Network methods share limitations

We compared the six network methods in a 5-fold subsampling setting (Section 2.5). In this comparison we measured four properties (Fig 3 and S4 Fig): the size of the solution; the sensitivity and the specificity of an L1-penalized logistic regression classifier on the selected SNPs; the stability of the methods; and their computational runtime. The solution size varied greatly between the different methods (Fig 3A). Heinz produced the smallest solutions, with an average of 182 selected SNPs (Section 2.3.5) while the largest ones came from SConES GI (6 256.6 SNPs) and dmGWAS (4 255.0 SNPs). LEAN did not produce any solution in any of the subsamples.

To determine whether the selected SNPs could predict cancer susceptibility, we computed the classifiers' performances on *test sets* (S4B Fig). The different classifiers displayed similarly low sensitivities and specificities, all in the 0.52—0.56 range. Interestingly, the classifier trained on all the SNPs had a similar performance, despite being the only method aiming to minimize prediction error. Of course, although these performances were low, we did not expect to separate cases from controls well using exclusively genetic data [53].

Another desirable quality of a selection algorithm is the stability of the solution with respect to small changes in the input (Section 2.5.1). Heinz was highly stable in our benchmark, while the other methods displayed similarly low stabilities (Fig 3B).

In terms of computational runtime, the fastest method was heinz (Fig 3C), which returned a solution in a few seconds. HotNet2 was the slowest (3 days and 14 hours on average). Including the time required to compute the gene scores, however, slowed down considerably gene-based methods; on this benchmark, that step took on average 1 day and 9.33 hours. Including this first step, it took 5 days on average for HotNet2 to produce a result.

Using different combinations of parameters (Section 2.3.4), we computed how good each of the methods was at recovering the results of a conventional GWAS on BCAC (Section 2.6, Fig 3D). SConES exhibits the largest area under the curve since, when $\lambda = 0$ (i.e., network topology is disregarded), it is equivalent to a Bonferroni correction. The remaining network methods have similar areas under the curve, with heinz having the largest one.

### 3.6 Network topology and association scores matter and might lead to ambiguous results

As shown above, and despite their similarities, the different ways of modeling the problem led to remarkably different solutions. Importantly, understanding which assumptions the methods made allowed us to understand the results more in depth. For instance, the fact that LEAN did not return any gene implied that there was no gene such that both itself and its environment were, on average, strongly associated with the disease.

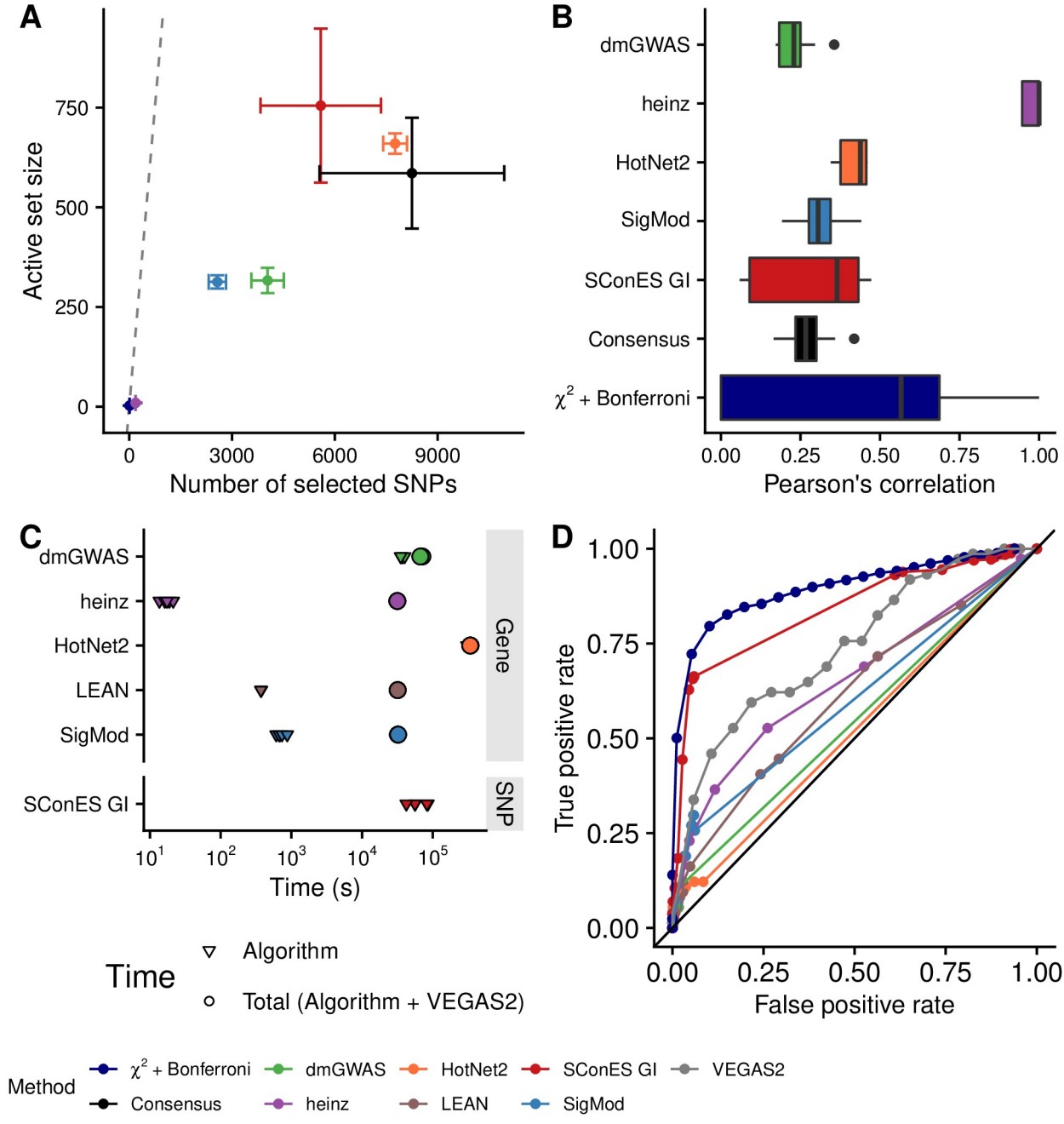

**Fig 3. Comparison of network methods on GENESIS.** Each method was run 5 times on a random subset containing 80% of the samples and tested on the remaining samples (Section 2.5). As LEAN did not select any gene, we excluded it from panels A and B. **(A)** Number of SNPs selected by each method and number of SNPs in the active set (i.e., the number of SNPs selected by the classifier, Section 2.5.2). Points are the average over the 5 runs; the error bars represent the standard error of the mean. A grey diagonal line with slope 1 is added for comparison, indicating the upper bound of the active set. For reference, the active set of the classifier using all the SNPs as input included, on average, 154 117.4 SNPs. **(B)** Pairwise Pearson correlations of the solutions produced by different methods. A Pearson correlation of 1 means the two solutions are the same. **(C)** Runtime of the evaluated methods, by type of network used (PPIN or SNP). For gene-based methods, inverted triangles represent the runtime of the algorithm alone, and circles the total time, which includes the algorithm themselves and the additional 119 980 seconds (1 day and 9.33 hours) that VEGAS2 took on average to compute the gene scores from SNP summary statistics. **(D)** True positive rate and false positive rate of the methods, obtained using different parameter combinations (Section 3.7). We used as true positives BCAC-significant SNPs (for SConES and $\chi^2$ + Bonferroni) and genes (for the remaining methods, Section 2.6). We used the whole dataset in this panel.

In the GENESIS dataset, heinz's solution was very conservative, providing a small solution with the lowest median P-value (Table 2). By repeatedly selecting this compact solution, heinz was the most stable method (Fig 3B). Its conservativeness stemmed from its preprocessing step, which modeled the gene P-values as a mixture model of a beta distribution and a uniform distribution, controlled by an FDR parameter. Due to the limited signal at the gene level in this dataset (S2B Fig), only 36 genes retained a positive score after that transformation. However, this small solution did not provide much insight into the susceptibility mechanisms to cancer. Importantly, it ignored genes that were associated with cancer in this dataset, like *FGFR2*.

On the other end of the spectrum, dmGWAS, HotNet2, and SigMod produced large solutions. DmGWAS' solution was the lowest scoring solution on average because of its greedy framework, which is biased towards larger solutions [54]. It considered all nodes at distance 2 of the examined subnetwork and accepted a weakly associated gene if it was linked to another, high scoring one. Aggregating the results of successive greedy searches exacerbates this bias, leading to a large, tightly connected cluster of unassociated genes (Fig 4A). This relatively low signal-to-noise ratio combined with the large solution requires additional analyses to draw conclusions, such as enrichment analyses. In the same line, HotNet2's solution was even harder to interpret, being composed of 440 genes divided into 135 subnetworks. Lastly, Sig-Mod missed some of the highest scoring breast cancer susceptibility genes in the dataset, like *FGFR2* and *TOX3*.

Another peculiarity of network methods was their relationship to betweenness centrality. We studied random rewirings of the PPIN that preserved node centrality (Section 2.7). In this setting, network methods favored central genes (Fig 4B) even though highly central genes often had no association to breast cancer susceptibility (Fig 4C). We found this bias especially in SigMod (S5 Fig), which selected three highly central, unassociated genes in both the PPIN and in many of the random rewirings: *COPS5*, *CUL3*, and *FN1*. However, as we showed in Section 3.3 and will show in 3.8, there is evidence in the literature of the contribution of the first two to breast cancer susceptibility. With regards to *FN1*, it encodes a fibronectin, a protein of the extracellular matrix involved in cell adhesion and migration. Overexpression of *FN1* has been observed in breast cancer [55], and it anticorrelates with poor prognosis in other cancer types [56, 57].

By using a SNP subnetwork, SConES analyzed each SNP in its functional context. Therefore, it could select SNPs located in genes not included in the PPIN and in non-coding regions. We compared the solution of SConES in the GI network (using PPIN information), to the one using only positional information (GS network) and to the one using positional and gene annotations (GM network). Importantly, SConES produced similar results on the GS and GM networks (S6 Fig). While the solutions on those two considerably overlap with SConES GI's, they contained additional gene-coding segments (Fig 4D). In fact, both SConES GS and GM selected chromosome regions related to breast cancer, like 3p24 (*SLC4A7/NEK10* [58]), 5p12 (*FGF10*, *MRPS30* [50]), 10q26 (*FGFR2*), and 16q12 (*TOX3*). In addition to those, SConES GS selected region 8q24, also linked to breast cancer (*POU5F1B* [59]).

## 3.7 Different parameters produce similarly-sized solutions

We explored methods' parameter space by running them under different combinations of parameters (Section 2.3.4). In agreement with their formulations (Section 2.3.3), larger values of specific parameters produced less stringent solutions (S7A Fig): for HotNet2 and heinz, this is the threshold above which genes receive a positive score; for dmGWAS, it is the d parameter, which controls how far neighbors could be added; for SigMod, it is nmax, which specifies the maximum size of the solution; and for LEAN, it is the P-value threshold to consider a solution

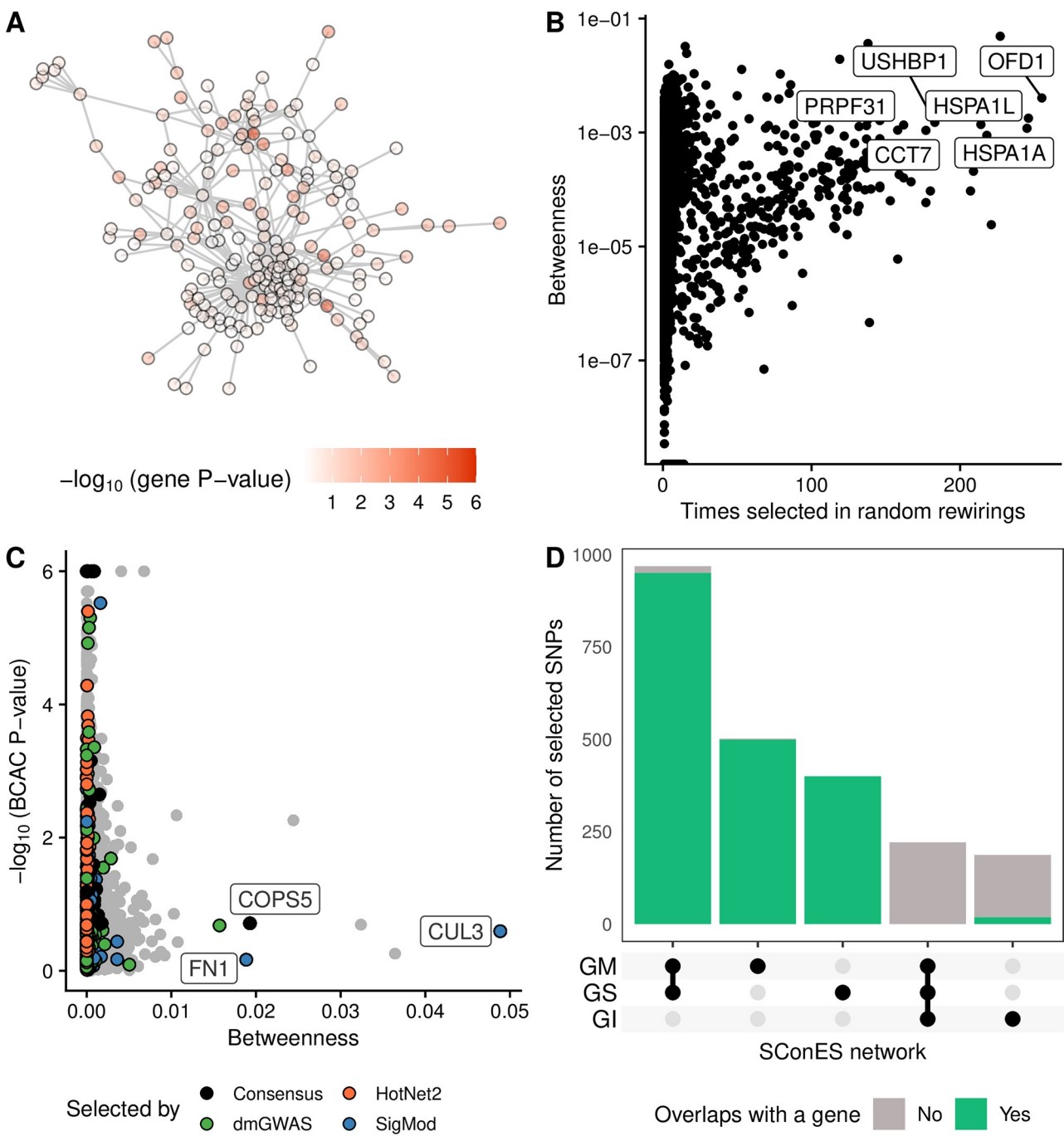

**Fig 4. Drawbacks encountered when using network methods. (A)** DmGWAS solution, with the genes colored according to the -log$_{10}$ of their P-value. **(B)** Number of times a gene was selected by either dmGWAS, heinz, LEAN, or SigMod in 100 rewirings of the PPIN (Section 2.4) and its betweenness. **(C)** Betweenness and -log$_{10}$ of the VEGAS2 P-value in BCAC for each of the nodes in the PPIN. We highlighted the genes selected by each method and the ones selected by more than one ("Consensus"). We labeled the three most central genes that were picked by any method. **(D)** Overlap between the solutions of SConES GS, GM, or GI. Barplots are colored based on whether the SNPs map to a gene or not (Section 2.3.5).

significant. Two parameters had the opposite effect (the larger, the more stringent): SigMod's `maxjump`, which sets the threshold to consider an increment in λ "large enough"; and SConES' $\eta$, where higher values produce smaller solutions. However, two of the parameters did not have the expected effect: dmGWAS' `r`, which controls the minimum increment in the

score required to add a gene; and SigMod's `maxjump`, which sets the threshold to consider an increment in λ "large enough". In both cases, the size of the solution was very similar across the different values. Despite the differences in size, the solutions' size was relatively robust to the choice of parameters (S7B Fig).

We computed the Pearson correlation between the different solutions as in Section 2.5.1 to study how the parameters affected which genes and SNPs were selected (S8 Fig). This analysis showed that dmGWAS and SigMod were robust to two parameters: the parameter `d` determined dmGWAS' output more than `r`; for SigMod, it was *nmax* rather than *maxjump*.

SConES presented an interesting case in terms of feature selection: most of the explored combinations of parameters led to trivial solutions (they included either all the SNPs or none of them) (S8 Fig). To explore a more meaningful parameter space, we selected the parameters in two rounds in our experiments. First, we explored the whole sample space. Then, we focused on a range of η and λ 1.5 orders of magnitude above and below the best parameters, respectively. This second parameter space was more diverse, which allowed to find more interesting solutions.

### 3.8 Building a stable consensus network preserves global network properties

Most network methods, including the consensus, were highly unstable (Fig 3B), raising questions about the results' reliability. We built a new, *stable consensus* solution using the genes selected most often across the 30 solutions obtained by running the 6 methods on 5 different splits of the data (Section 2.5). Such a network should capture the subnetworks more often found altered, and hence should be more resistant to noise. We used only genes selected in at least 7 solutions, which corresponded to 1% of all genes selected at least once. The resulting stability-based consensus was composed of 68 genes (Fig 5A). This network shared most of the properties of the consensus: breast cancer susceptibility genes were overrepresented (P-value = $3 \times 10^{-4}$), as well as genes involved in mitochondrial translation and the attenuation phase (adjusted P-values 0.001 and $3 \times 10^{-5}$ respectively); the selected genes were more central than average (P-value = $1.1 \times 10^{-14}$); and a considerable number of nodes (19) were isolated (Fig 5B and 5C).

Despite these similarities, the consensus and the stable consensus included different genes. In the stable consensus network, the most central gene was *CUL3*, which was absent from the previous consensus solution and had a low association score in both GENESIS and BCAC (P-values of 0.04 and 0.26, respectively). This gene is a component of Cullin-RING ubiquitin ligases. It impacts the protein levels of multiple genes relevant for cancer progression [60], and its overexpression has also been linked to increased sensitivity to carcinogens [61].

## 4 Discussion

In recent years, the GWAS' ability to unravel the mechanisms leading to complex diseases has been called into question [6]. First, the omnigenic model proposes that gene functions are interwoven in a dense co-function network. The practical consequence is that larger and larger GWAS will lead to discovering an uninformative wide-spread pleiotropy. Second, its conservative statistical framework hinders GWAS discovery. Network methods elegantly address these two issues by using both association scores and an interaction network to consider the biological context of each of the genes and SNPs. Based on what could be considered diverse interpretations of the omnigenic model, several methods for network-guided discovery have been proposed in recent years. In this article we evaluated six of these methods (Section 2.3.3) by applying them to the GENESIS GWAS dataset on familial breast cancer (Section 2.1).

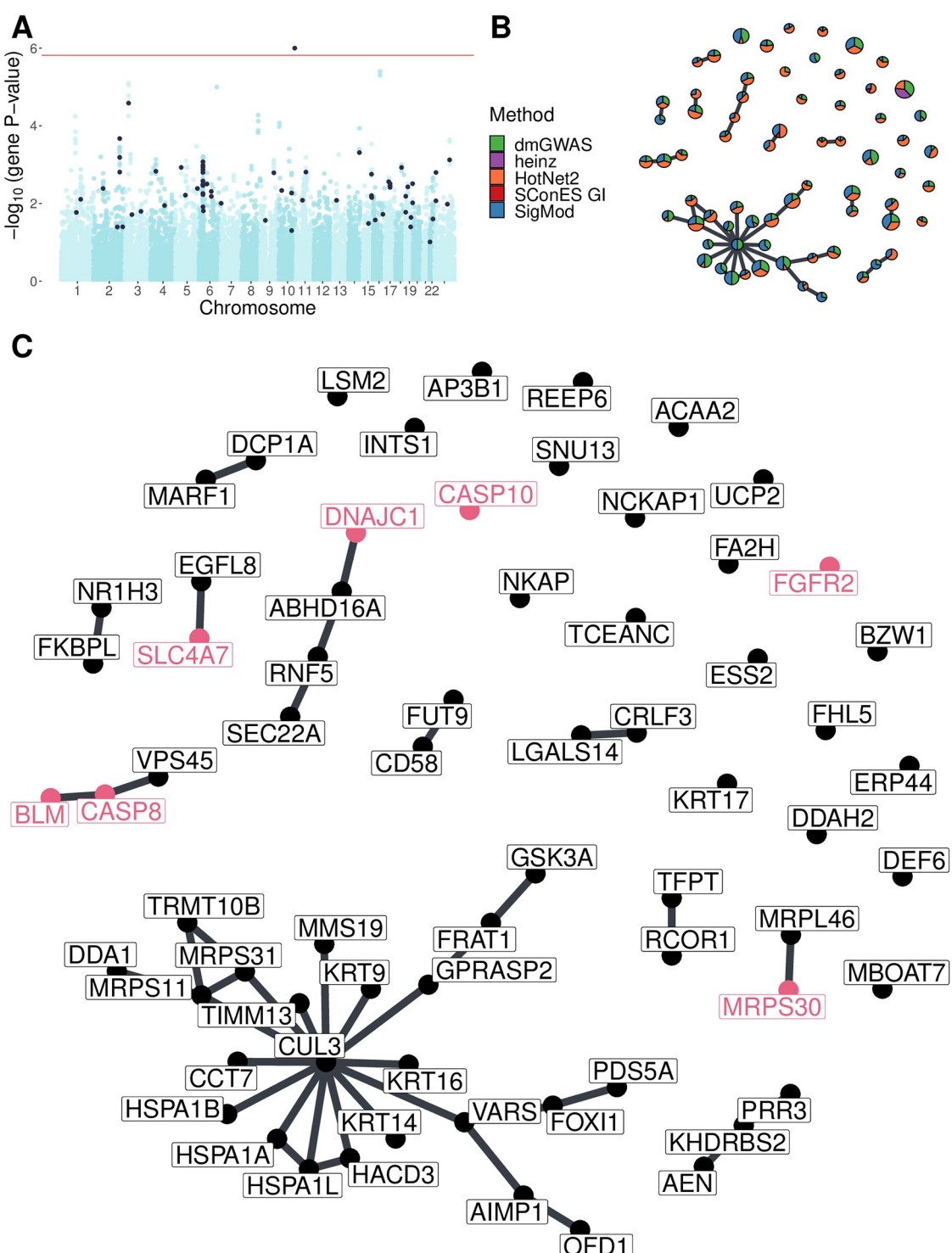

**Fig 5. Stable consensus solution on GENESIS (Section 3.8). (A)** Manhattan plot of genes; in black, the ones in the stable consensus solution. The red line indicates the Bonferroni threshold ($1.53 \times 10^{-6}$ for genes). **(B)** Stable consensus network. Each gene is represented by a pie chart, which shows the methods that selected it. We enlarged the most central gene (*CUL3*), the known breast cancer susceptibility genes, and the BCAC-significant genes (Section 2.6). **(C)** The nodes are in the same disposition as in panel B, but we indicated every gene name. We colored in pink the names of known breast cancer susceptibility genes and BCAC-significant genes.

DmGWAS, Heinz, HotNet2, SConES, and SigMod all yielded compelling solutions, which include (but are not limited to) known breast cancer susceptibility genes (Section 3.2). In general, the selected genes and SNPs were more central than most other genes and SNPs, agreeing with the observation that disease genes are more central [49]. However, very central nodes are also more likely to be connecting any given random pair of nodes, making them more likely to be selected by network methods (Section 3.6). However, we found support in the literature for the involvement of the selected highly central genes (*COPS5*, *FN1*, and *CUL3*) in breast cancer susceptibility (Sections 3.3, 3.6, and 3.8). Despite these similarities, the methods' solutions were notably different. At one end of the spectrum, SConES and heinz preferred high scoring solutions, which were also small and hence did not shed much light on the disease's etiology. On the other end, dmGWAS, HotNet2 and SigMod gravitated towards lower scoring but larger solutions, which provided a wide overview of the biological context. While this deepens our understanding of breast cancer susceptibility and provides biological hypotheses, interpreting their solutions required further analyses. For instance, we examined the centrality of the selected genes to understand how much that property was driving their selection (Section 3.6). However, all solutions shared two drawbacks. First, they were all equally bad at discriminating cases from controls. However, the classification accuracy of network methods was similar to that of a classifier trained on the entire genome (Section 3.5), which suggests that cases and controls are difficult to separate in the GENESIS dataset. This may be due to limited statistical power, which reduces the ability to identify relevant SNPs. However, in any event, we do not expect to separate people who have or will develop cancer from others on the sole basis of their genomes, ignoring all environmental factors and chance events. Hence, network methods were preferable to the logistic regression classifier since they did "no worse" at classification while providing an interpretable solution. Using an algorithm other than L1-penalized logistic regression might improve classification accuracy. Specifically, PRS-based strategies seem to perform slightly better on GENESIS [62]. Second, all methods were remarkably unstable, yielding different solutions for slightly different inputs. This might partly have been caused by the instability of the P-values themselves in low statistical power settings [63]. Hence, heinz's conservative transformation of P-values, which favored only the most extreme ones, led to improved stability. Another source of instability might have been the redundancy inherent to biological networks, a consequence of an evolutionary pressure to avoid single points of failure [64]. Hence, biological networks will often have multiple paths connecting two high-scoring nodes.

To overcome these limitations while exploiting each method's strengths, we proposed combining them into a consensus solution. We used the straightforward strategy of including any node that was recovered by at least two methods. We thus proposed two solutions (Sections 3.3 and 3.8): a consensus solution, which addressed the heterogeneity of the solutions, and a stable consensus solution, which addresseded the instability of the methods. They both included the majority of the strongly associated smaller solutions and captured genes and broader mechanisms related to cancer, thus synthesizing the mechanisms altered in breast cancer cases. Thanks to their smaller size and network structure, they provided compelling hypotheses on genes like *COPS5* and *CUL3*, which lack genome-wide association with the disease but are related to cancer at the expression level and consistently interact with high scoring genes. Notably, while the consensus approach was as unstable as the individual network-guided methods, the stable consensus network retained the ability to provide compelling hypotheses and had better stability. This supported that redundant but equivalent mechanisms might cause instability and supported the conclusions obtained on the individual solutions.

In this work, we have compared our results to significant genes and SNPs in the BCAC study [42]. Network methods showed modest precision but much higher recall at recovering

BCAC hits (Section 3.4). While precision might be desirable when a subset of useful markers is required (for instance, for diagnosis), higher recall is desirable in exploratory settings. Nonetheless, BCAC was not an ideal ground truth. First, the studied populations are not entirely overlapping: BCAC focused on a pan-European cohort, while GENESIS targeted the French population. Second, the study designs differed: a high proportion of breast cancer cases investigated in BCAC were sporadic (not selected according to family history), while GENESIS was a homogeneous dataset not included in BCAC focused on the French high-risk population attending the family cancer clinics. Finally, and this is indeed the motivation for this study, GWAS are unlikely to identify all genes relevant for the disease: some might only show up in rare-variant studies; others might have too small effect sizes. Network methods account for this by including genes with low association scores but with relevant topological properties. Hence, network methods and GWAS, even when well-powered, are unlikely to capture exactly the same sets of genes. This might partly excuse the low precisions displayed in Section 3.4 and the low AUC displayed in Section 3.5.

As not all PPIN databases compiled the same interactions, the choice of the PPIN determines the final output. In this work, we used only interactions from HINT from high-throughput experiments (Section 2.3.2). This responded to concerns about adding interactions identified in targeted studies and falling into "rich getting richer" problems: since popular genes have a higher proportion of their interactions described [10, 24], they might bias discovery towards themselves by reducing the average shortest path length between two random nodes. On the other hand, Huang et al. [12] found that larger networks were more useful than smaller networks to identify disease genes. This would support using the largest networks in our experiments. However, when we compared the impact of using a larger PPIN containing interactions from both high-throughput experiments and the literature (Section 2.3.2), for most of the methods it did not change much the size or the stability of the solution, the classification accuracy, or the runtime (S9 Fig). This supports using only interactions from high-throughput experiments, which produced similar solutions and avoided falling into "circular reasonings", where the best-known genes were artificially pushed into the solutions, as we observed in Section 3.6.

The strength of network-based analyses comes from leveraging prior knowledge to boost discovery. In consequence, they show their shortcomings on understudied genes, especially those not in the network. Out of the 32 767 genes to which we mapped the genotyped SNPs, 60.7% (19 887) were not in the PPIN. The majority of those (14 660) are non-coding genes, mainly lncRNA, miRNA, and snRNA (S10 Fig). Nevertheless, RNA genes like *CASC16* were associated to breast cancer (Section 3.1), reminding us of the importance of using networks beyond coding genes. Besides, even protein-coding genes linked to breast cancer susceptibility [58], like *NEK10* (P-value $1.6 \times 10^{-5}$, overlapping with *SLC4A7*) or *POU5F1B*, were absent from the PPIN. However, on average protein-coding genes absent from the PPIN were less associated with breast cancer susceptibility (Wilcoxon rank-sum P-value = $2.79 \times 10^{-8}$, median P-values of 0.43 and 0.47). This could not be due to well-known genes having more known interactions because we only used interactions from high-throughput experiments. As disease genes tend to be more central [49], we hypothesize that it was due to interactions between central genes being more likely. It is worth noting that network methods that do not use PPIs, like SConES GS and GM, recovered SNPs in *NEK10* and *CASC16*. Moreover, both SConES GM and GI recovered intergenic regions, which might contain key regulatory elements [65], but are excluded from gene-centric approaches. This shows the potential of SNP networks, in which SNPs are linked when there is evidence of co-function, to perform network-guided GWAS even in the absence of gene-level interactions. Lastly, all the methods are heavily

affected by how SNPs are mapped to genes, and other strategies (e.g., eQTLs, SNPs associated to gene expression) might lead to different results.

A crucial step for the gene-based methods is the computation of gene scores. In this work, we used VEGAS2 [22] due to the flexibility it offers to use user-specified gene annotations. However, it presents known problems: selection of an appropriate percentage of top SNPs, long runtimes and P-value precision limited to the number of permutations [29]). Additionally, other algorithms like PEGASUS [29], SKAT [66] or COMBAT [67] might have more statistical power.

How to handle linkage disequilibrium (LD) is often a concern among GWAS practitioners. Often, the question is whether an LD-based pruning of the analyzed SNPs will improve the results. VEGAS2 accounts for LD patterns, and hence an LD pruning step would not impact gene-based network methods, although it would speed up VEGAS2's computation time. In Section 3.3 we highlighted ambiguities that appear when genes overlap or contain SNPs that are in LD. The presented case is paradigmatic since all three genes are in the HLA region, the most gene-dense region of the genome [68]. Network methods are prone to selecting such genes when they are functionally related, and hence interconnected in the PPIN. But the opposite case is also true: when genes are not functionally related (and hence disconnected in the PPIN), network methods might disregard them even if they have high association scores. With regards to SConES, fewer SNPs would lead to simpler SNP networks and, possibly, shorter runtimes. However, LD patterns also affect SConES' in other ways, since its formulation penalizes selecting a SNP and not its neighbors, via a nonzero parameter $\eta$ in Eq 5. Due to LD, nearby SNPs' P-values correlate; since positional information determines SNP networks, nearby SNPs are likely to be connected. Hence, SConES tends to select LD-blocks formed by low P-value SNPs. This might explain why SConES produced similar results on the GS and GM networks, heavily affected by LD (Section 3.6). However, this same behavior raises the burden of proof required to select SNPs with many interactions, like those mapped to hub genes in the PPIN. For this reason, SConES GI did not select any protein coding gene. This could be caused by the absence of joint association of a gene and most of its neighbors, a hypothesis supported by LEAN's lack of results. Yet, a different combination of parameters could lead to a more informative SConES' solution (e.g., a lower $\lambda$ in Eq 5), although it is unclear how to find it. In addition, due to the design of the iCOGS array (Section 2.1), the genome of GENESIS participants has not been unbiasedly surveyed: some regions are finemapped—which might distort gene structure in GM and GI networks—while others are understudied—hindering the accuracy with which the GS network captures the genome structure. A strong LD pruning might address such problems.

To produce the two consensus solutions, we faced practical challenges due to the differences in interfaces, preprocessing steps, and unexpected behaviors of the various methods. To make it easier for others to apply these methods to new datasets and aggregate their solutions, we built six `nextflow` pipelines [69] with a consistent interface and, whenever possible, parallelized computation. They are available on GitHub: hclimente/gwas-tools (Section 2.9). Importantly, we compiled those methods with a permissive license into a Docker image for easier use, available on Docker Hub hclimente/gwas-tools.

## Supporting information

**S1 Table. SNP summary statistics on GENESIS.**
(TSV)

**S2 Table. Gene summary statistics on GENESIS.**
(TSV)

**S3 Table. Summary statistics on the results of SConES on the three SNP networks (Section 2.3.2).** The first row within each block contains the summary statistics on the whole network.
(PDF)

**S4 Table. Pathway enrichment analyses of the genes in SigMod's solution.**
(TSV)

**S5 Table. Pathway enrichment analyses of the genes in dmGWAS' solution.**
(TSV)

**S6 Table. Pathway enrichment analyses of the genes in HotNet2's solution.**
(TSV)

**S7 Table. Pathway enrichment analyses of the genes in the consensus' solution.**
(TSV)

**S1 Fig. GENESIS shows no differential population structure between cases and controls. (A,B,C,D)** Eight main principal components, computed on the genotypes of GENESIS. Cases are colored in green, controls in orange.
(TIF)

**S2 Fig. Association in GENESIS. The red lines represent the Bonferroni thresholds. (A)** SNP association, measured from the outcome of a 1 d.f. $\chi^2$ allelic test (Section 2.2). Significant SNPs within a coding gene, or within 50 kilobases of its boundaries, are annotated. The Bonferroni threshold is $2.54 \times 10^{-7}$. **(B)** Gene association, measured by P-value of VEGAS2 [22] using the 10% of SNPs with the lowest P-values (Section 2.2). The Bonferroni threshold is $1.53 \times 10^{-6}$. **(C)** SNP association as in panel (A). The SNPs in black were selected by an L1-penalized logistic regression (Section 2.5.2, $\lambda = 0.03$).
(TIF)

**S3 Fig. Relationship between the $\log_{10}$ of the betweenness centrality and the $-\log_{10}$ of the VEGAS2 P-value of the genes in the consensus solution.** The blue line represents a fitted generalized linear model.
(TIF)

**S4 Fig. Additional benchmarks of the network methods. (A)** Precision and recall of the evaluated methods with respect to Bonferroni-significant SNPs/genes in BCAC. For reference, we added a gray line with a slope of 1. **(B)** Sensitivity and specificity on the test set of the L1-penalized logistic regression trained on the features selected by each of the methods. The performance of the classifier trained on all SNPs is also displayed. Points are the average over the 5 runs; the error bars represent the standard error of the mean.
(TIF)

**S5 Fig. Number of times a gene was selected by either dmGWAS, heinz, or SigMod in 100 rewirings of the PPIN (Section 2.7) and its betweenness.** This figure is equivalent to Fig 4B, split by method.
(TIF)

**S6 Fig. Pearson correlation between the different solutions. (A)** Correlation between selected SNPs. **(B)** Correlation between selected genes. In general, the solutions display a very low overlap.
(TIF)

**S7 Fig. Size of the solutions obtained under different parameters. (A)** Size of the solution produced by different parameter values, expressed as a percentage of the maximum solution size for the method, or the highest tested value for the parameter, respectively. The size of the solution is the median among all the solution sizes for the same parameter. **(B)** Boxplot of the solution sizes of the methods under the explored parameters (Section 2.3.4).
(TIF)

**S8 Fig. Pearson correlation between the solutions obtained under different parameters, computed as in Section 2.5.1.** Grey tiles represent the cases where we could not compute the Pearson correlation because the two vectors were either all ones (all genes/SNPs were selected) or zeros (no genes/SNPs were selected).
(TIF)

**S9 Fig. Comparison of the benchmark on high-throughput (HT) interactions to the benchmark on both high-throughput and literature curated interactions (HT+LC).** Grey lines represent no change in the statistic between the benchmarks (1 for ratios mean(HT) / mean (HT + LC), 0 for differences mean(HT)—mean(HT + LC)). **(A)** Ratios of the selected features between both benchmarks and of the active set (Section 2.5.2). **(B)** Shifts in sensitivity and specificity. **(C)** Shift in Pearson correlation between benchmarks. **(D)** Ratio between the runtimes of the benchmarks. For gene-based methods, inverted triangles represent the ratio of runtimes of the algorithms themselves, and circles the total time, which includes the algorithm themselves and the additional 119 980 seconds (1 day and 9.33 hours) that VEGAS2 took on average to compute the gene scores from SNP summary statistics. In general, adding additional interactions slightly improved the stability of the solution. However, it increased the solution size and the required runtime, and had mixed effects on the sensitivity and specificity.
(TIF)

**S10 Fig. Biotypes of genes from the annotation that are not present in the HINT PPIN.**
(TIF)

## Acknowledgments

We wish to thank Om Kulkarni for helpful discussion on gene-based GWAS and PPIN databases, Vivien Goepp for his constructive comments of the manuscript, and the genetic epidemiology platform (the PIGE, Plateforme d'Investigation en Génétique et Epidemiologie: S. Eon-Marchais, M. Marcou, D. Le Gal, L. Toulemonde, J. Beauvallet, N. Mebirouk, E. Cavaciuti), the biological resource center (S. Mazoyer, F. Damiola, L. Barjhoux, C. Verny-Pierre, V. Sornin).

We thank all the GENESIS collaborating cancer clinics (Clinique Sainte Catherine, Avignon: H. Dreyfus; Hôpital Saint Jacques, Besançon: M-A. Collonge-Rame; Institut Bergonié, Bordeaux: M. Longy, A. Floquet, E. Barouk-Simonet; CHU, Brest: S. Audebert; Centre François Baclesse, Caen: P. Berthet; Hôpital Dieu, Chambéry: S. Fert-Ferrer; Centre Jean Perrin, Clermont-Ferrand: Y-J. Bignon; Hôpital Pasteur, Colmar: J-M. Limacher; Hôpital d'Enfants CHU—Centre Georges François Leclerc, Dijon: L. Faivre-Olivier; CHU, Fort de France: O. Bera; CHU Albert Michallon, Grenoble: D. Leroux; Hôpital Flaubert, Le Havre: V. Layet; Centre Oscar Lambret, Lille: P. Vennin, C. Adenis; Hôpital Jeanne de Flandre, Lille: S. Lejeune-Dumoulin, S. Manouvier-Hanu; CHRU Dupuytren, Limoges: L. Venat-Bouvet; Centre Léon Bérard, Lyon: C. Lasset, V. Bonadona; Hôpital Edouard Herriot, Lyon: S. Giraud; Institut Paoli-Calmettes, Marseille: F. Eisinger, L. Huiart; Centre Val d'Aurelle—Paul Lamarque, Montpellier: I. Coupier; CHU Arnaud de Villeneuve, Montpellier: I. Coupier, P. Pujol; Centre

René Gauducheau, Nantes: C. Delnatte; Centre Catherine de Sienne, Nantes: A. Lortholary; Centre Antoine Lacassagne, Nice: M. Frénay, V. Mari; Hôpital Caremeau, Nîmes: J. Chiesa; Réseau Oncogénétique Poitou Charente, Niort: P. Gesta; Institut Curie, Paris: D. Stoppa-Lyonnet, M. Gauthier-Villars, B. Buecher, A. de Pauw, C. Abadie, M. Belotti; Hôpital Saint-Louis, Paris: O. Cohen-Haguenauer; Centre Viggo-Petersen, Paris: F. Cornélis; Hôpital Tenon, Paris: A. Fajac; GH Pitié Salpétrière et Hôpital Beaujon, Paris: C. Colas, F. Soubrier, P. Hammel, A. Fajac; Institut Jean Godinot, Reims: C. Penet, T. D. Nguyen; Polyclinique Courlancy, Reims: L. Demange*, C. Penet; Centre Eugène Marquis, Rennes: C. Dugast*; Centre Henri Becquerel, Rouen: A. Chevrier, T. Frebourg, J. Tinat, I. Tennevet, A. Rossi; Hôpital René Huguenin/Institut Curie, Saint Cloud: C. Noguès, L. Demange*, E. Mouret-Fourme; CHU, Saint-Etienne: F. Prieur; Centre Paul Strauss, Strasbourg: J-P. Fricker, H. Schuster; Hôpital Civil, Strasbourg: O. Caron, C. Maugard; Institut Claudius Regaud, Toulouse: L. Gladieff, V. Feillel; Hôpital Bretonneau, Tours: I. Mortemousque; Centre Alexis Vautrin, Vandoeuvre-les-Nancy: E. Luporsi; Hôpital de Bravois, Vandoeuvre-les-Nancy: P. Jonveaux; Gustave Roussy, Villejuif: A. Chompret*, O. Caron). *Deceased prematurely.

## Author Contributions

**Conceptualization:** Héctor Climente-González, Christine Lonjou, Chloé-Agathe Azencott.

**Data curation:** Christine Lonjou.

**Formal analysis:** Héctor Climente-González, Christine Lonjou.

**Funding acquisition:** Dominique Stoppa-Lyonnet, Nadine Andrieu, Chloé-Agathe Azencott.

**Investigation:** Héctor Climente-González, Christine Lonjou.

**Methodology:** Héctor Climente-González, Christine Lonjou, Chloé-Agathe Azencott.

**Project administration:** Chloé-Agathe Azencott.

**Resources:** Dominique Stoppa-Lyonnet, Nadine Andrieu.

**Software:** Héctor Climente-González, Christine Lonjou.

**Supervision:** Christine Lonjou, Fabienne Lesueur, Nadine Andrieu, Chloé-Agathe Azencott.

**Validation:** Christine Lonjou, Fabienne Lesueur.

**Visualization:** Héctor Climente-González.

**Writing – original draft:** Héctor Climente-González.

**Writing – review & editing:** Héctor Climente-González, Christine Lonjou, Fabienne Lesueur, Nadine Andrieu, Chloé-Agathe Azencott.

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
