## [Decision Letter · Decision Letter 0]

18 Aug 2020

Dear %TITLE% Climente-González,

Thank you very much for submitting your manuscript "Comparing and combining network-guided biomarker discovery to discover genetic susceptibility mechanisms to breast cancer on the GENESIS study" for consideration at PLOS Computational Biology.

As with all papers reviewed by the journal, your manuscript was reviewed by members of the editorial board and by several independent reviewers. In light of the reviews (below this email), we would like to invite the resubmission of a significantly-revised version that takes into account the reviewers' comments.

We cannot make any decision about publication until we have seen the revised manuscript and your response to the reviewers' comments. Your revised manuscript is also likely to be sent to reviewers for further evaluation.

Sincerely,

Sushmita Roy, Ph.D.

Associate Editor

PLOS Computational Biology

Florian Markowetz

Deputy Editor

PLOS Computational Biology

Reviewer's Responses to Questions

**Comments to the Authors:**

Reviewer #1: This manuscript takes a systems biology approach to analyze the GENESIS data set of breast cancer GWAS. Overall, six network-based approaches were compared. A consensus network was created by taking all nodes identified by at least 2 network algorithms. With so many competing computational methods that seek to solve the same problem, comparing the approaches on the same data set provides a helpful perspective. However, the manuscript does not provide an in depth comparison of the methodologies, does not perform parameter optimization for each methodology, and does not provide comprehensive documentation of the results.

Major revisions:

* Overall, the research question and gap that this manuscript seeks to fill is not clearly articulated. No new novel computational methodologies are presented and the results read more as a case report/review article. In addition, the results from each of the network approaches are presented, but no comparison or contrasts are fully discussed. This is a missed opportunity for understanding the advantages of disadvantages of the methodologies.

* The introduction was non-specific. If the goal of the manuscript is to highlight the challenges of identifying SNPs in breast cancer, then the introduction should highlight the prior studies that have attempted to identify SNPs. If the goal of the manuscript is to focus on analytical methods, then more discussion of other approaches for analyzing GWAS data is necessary.

* The methods and results lacked reason and justification for decisions on parameter values. Selection of parameters could impact the final results substantially. Since the goal of the manuscript is to compare and contrast different network approaches, more exploration of the parameter space of each methodology is needed.

* It is unclear what information was provided to each of the network methods. Specifically, all but one network algorithm uses gene level values such as p-values for each gene. The approach to merging SNP level data to gene level data for the network methods is not described.

* The precision/recall of these methodologies compared to BCAC are underwhelming. It is more preferable to have high precision or high recall?

* The pathway analysis methods results were under-reported. This analysis was performed for each network algorithm result and the consensus network. No full pathway analysis results are present in either the main text or supplemental document. The network algorithm results were deemed “similar”, but this cannot be confirmed. Moreover, the results were manually curated with no criteria for such curation. Were pathways selected because they were supported by literature or because they represented common themes?

* The consensus was determined by taking the set of nodes that were present in at least two of the six methods. How would the results vary if a different number of method identification were used? Also, there is no description how the edges were identified in the consensus network.

Minor revisions:

* The figures are referenced out of order.

* “Heinz produced the smallest solutions, with an average of 182 selected SNPs”. It is unclear how the gene-level network methods select SNPs.

* How are Pearson’s correlation of a solution set computed.

* What were the computing resources available on the machine used for testing the algorithm runtime?

Reviewer #2: The paper entitled “Comparing and combining network-guided biomarker discovery to discover genetic susceptibility mechanisms to breast cancer on the GENESIS study” proposes the implementation and comparison of different algorithms to identify subnetworks of interest from the integration of network and GWAS data. These applications are done in the particular context of the GENESIS study, a french GWAS of patients without BRCA1/2 pathogenic variants.

Overall, the approach and results are interesting, in particular the comparison of the outputs of different algorithms used to leverage network data are of interest to the community. However, despite this relevant content, the paper suffers from style and organisation weaknesses making it difficult to follow.

General comments:

Take care of using the same word(s) when referring to a unique concept. For example, “stable consensus” and “stable consensus subnetwork” are both used, or “molecules”, “biomolecules” and “genes”. Other similar examples of unnecessary use of synonyms are “PPIN”, “protein-protein interaction network” “gene-gene interactions” and “network”, and “solution network”, “causal network”, “causal subnetwork”, ”subnetworks”, “susceptibility network”. Similarly, please use the same formulation to refer to the 3 breast cancer datasets along the text. Overall, the manuscript would highly benefit from a simplification of the text.

The style is sometimes unnecessarily complicated, please simplify.

The reference numbering does not start at [1]. The figure numbering does not start at 1 in the text, and does not follow the order of quotation of the figures. For instance, in the results section you start referring to your supplementary materials with the Supplementary Figure 2A whereas you haven’t mentioned the Supplementary Figure 1. Figure 3B and 1B are quoted before Figure 1A, Figure 1C is referred after Figure2...

Overall, the referencing organisation of the paper lacks clarity. Aside from the reference and figure numbering, the method section is not quoted when needed. For instance, the results start by quoting Section 4.3.1 in the first sentence, but it’s not mentioned that this corresponds to the method section, and the previous sections of the method section, explaining the first parts of the analyses, are not quoted in the results. Please use consistent and logical organisation, in particular to quote the method section.

Several acronyms are used for the first time without a definition (e.g. PPIN and BH).

The network figures 2 and 5 could be improved for readability and for information content. Please consider improving the visibility of the nodes and edges, removing the lines pointing at the gene labels because it looks like additional edges. In addition, having only the names of some of the nodes lowers the interest of the figures. The networks are not so big, and a software such as cytoscape should allow to display both node pie charts and all node labels on the figure.

Add numbers to all the equations, only one equation is currently numbered

Be consistent with the use of tenses. For instance, in section 4.3.2 you speak in present tense, but in section 4.3.3 you speak in past tense.

The global organisation of the figures might be redesigned, as it does not seem logical nor aligned with the flow in the text. For instance, figure 1D could go with figure 2, Figure 1C could go with figure 3 … The current flow of the text is scattered in the different figures, please consider redesigning the figures to follow the main ideas and sections.

Many formulations and sentences could be more specific, such as “we show a general alteration of the neighborhood…”, “significantly large overlap...”, “larger subnetworks...”, “it is located near...”.

Detailed comments on the text

Abstract

The sentence “By trading statistical stringency for biological meaningfulness, most network methods give more compelling results than standard SNP- and gene-level analyses, recovering causal subnetworks tightly related to cancer susceptibility.” is not clear, consider rephrasing. In addition, do the methods really trade statistical stringency? They just use other types of statistical tests, which are also selected for significance.

In the sentence “Yet, network methods are notably unstable, producing different results when the input data changes slightly.”, what do you mean by “slightly”? Does it apply to the network, the GWAS results or to both? Is this sentence a result of the study or a statement from the background knowledge?

“More central positions in the network than average” => than average what ?

Author summary

“pipeline” is not clearly defined

Introduction

Syntax: “Bonferroni correction is overly conservative when the statistical tests are correlated, as it* is the case in GWAS”.

List a couple of such properties in the sentence “In fact, an examination of biological networks shows that disease genes have differential properties.”.

The sentence “Network-based biomarker discovery methods exploit this relatedness to identify disease genes on GWAS data” is not clear, consider rephrasing. What does “relatedness” mean here? “Such is the extreme case of LEAN [25], which focuses on “star” subnetworks, i.e. instances were both a gene and its direct interactors are associated with the disease.” -> does this apply to all the neighbors?

The sentence “While all of them capture susceptibility mechanisms resembling that postulated by the omnigenic model, they do so in different ways, and provide a representative view of the field.” is not clear, consider rephrasing.

The introduction states a lot of concepts that could be defined more precisely to avoid confusion. For instance, a proper definition of biomarker could help as it can be seen as all the nodes present in a discovered subnetwork, or only the nodes with a significant p-value, or as cancer drivers, for instance. These concepts of significant genes, cancer drivers, biomarkers … are used without proper definitions.

Results

Section 2.1

“Moreover, and in opposition to what would be expected under the omnigenic model … are not interconnected in the PPIN” => I’m not sure the omnigenic model would lead to this statement, in particular as it is defined in the introduction. This is more a statement of network biology.

The sentence “In addition, the classification performance of the method is very low, and L1-penalized methods select only one of several correlated variables and are prone to instability, which further complicates interpretation.” is not clear, consider rephrasing.

In the sentence “This motivates exploring network methods, which trade statistical significance for biological relevance to find susceptibility subnetworks.”, what does “susceptibility subnetworks” mean? Please consider this comment in light of the general comment about the use of synonyms to refer to the same concept. In addition, there is a discrepancy in the organisation of the arguments in this section, because it states “the genes to which these SNPs map to are not interconnected in the PPIN” and “This motivates exploring network methods”.

Consider rephrasing the sentence “In fact, such methods provided comparably (poor) classification performance to L1-penalized logistic regression (Figure 3B), while providing more interpretable solutions.” to state more precisely the underlying idea, as bad results could be considered as bad results and thereby cannot really be interpreted.

Section 2.2

It’s not clear how the overlap is calculated with a Pearson correlation. Why not use Jaccard similarity or adjusted Rand Index?

LEAN is described as not retrieving “networks” in the text, and “genes” in the figure legend.

“While SConES GI failed to recover any protein coding gene...” -> is the goal to retrieve protein coding genes?

Figure 1.

(C) Put the names of the methods as legend, to improve readability. This could also be done for the similar pictures in Figure 3.

(D) The consensus network is the same as in Figure 2. Please consider fusing these Figures.

Bonferroni thresholds mentioned here in the legend but not in the method section

Table 1.

Add the number of connected components.

In the sentence “Lastly, a pathway enrichment analysis (Section 4.4) also showed similarities and differences in the underlying mechanisms.”, are you talking about all the methods or only about HotNet2 results? Underlying mechanisms of what? Not clear, consider rephrasing.

Section 2.3

“Despite the heterogeneity of the solutions, their shared properties suggest that each method captures different aspects” => this seems self-contradictory

In the sentence “we built a consensus subnetwork that captures the mechanisms most shared among the solution subnetworks (Section 4.3.4).”, how do you define a mechanism?

What is the overlap between at least 3 methods? And between all the methods?

Which tool did you use to do the pathway enrichment analysis?

Why do you refer to VEGAS2 p-values in this section and BH or Bonferonni in the previous ones? Do all these different words refer to different experiments?

“The opposite” => to what?

“This suggests that they were selected” => “they” refer to what?

“Despite its lack of association … its neighbors in the consensus …” => and in the PPIN?

Section 2.4

This section starts with the precise description of the BCAC dataset, which has been mentioned already, for instance in the legend of Figure 2. Please consider a reorganisation to give the information when needed first.

Typo “...at which most network methods operate.”

What is the correlation between BCAC and GENESIS at the gene level?

Section 2.5

Add the characteristics of the computer where you run the experiments in the method section.

Why “we do not expect to separate cases from controls well using exclusively genetic data”?

Section 2.6

The sentence “This is due to the particularities of each method, and directly or indirectly provides information about the dataset.” is unclear, consider rephrasing.

Figure 3.

Add the legend in an independent box to avoid overlapping text in (A) and (B).

The methods in (C) and (D) are listed in different order.

(D) The inverted triangle of SConES GI is missing (or not visible). Why is there a blank space between the legend “Gene” and “SNP”?

What “conventional GWAS” refers to? What is the difference with “Genesis-significant” used in Figure 1?

What is a solution “used”, do you mean produced?

What do you mean by “active set”?

Which lines represent the standard error of the mean?

Figure 4.

Why did you use a threshold of P-value < 0.1? Does this correspond to the GWAS p-values?

Figure 4C is not clearly explained.

Figure 4B, why is this computed on BCAC and not on GENESIS data?

Figure 4A, why not using a color scale to display the significant p-values, instead of significant/non-significant?

Add the name of the dataset in the sentence “In this dataset, heinz’s solution is very conservative...”.

In the sentence “Due to this parsimonious and highly associated solution, it was the best method to stably select a set of biomarkers (Figure 3C).”, what do you mean by “parsimonious”? Highly associated with what?

In the paragraph starting with the sentence “On the other end of the spectrum, dmGWAS, HotNet2, and SigMod produced large solutions.”, the use of the terms such as “least associated” and “weakly associated” is not clear.

The sentence “In fact, due to linkage disequilibrium, SConES favors such genes, as selecting SNPs in an LD-block which overlaps with a gene favors selecting the rest of the gene.” is unclear, consider rephrasing.

In the sentence “This makes it conservative regarding SNPs with many interactions, like those mapped to hub genes in the PPIN.”, what does “conservative” mean?

Define iCOGS array.

The end of the section 2.6 might belong to the discussion

Section 2.7

The sentence “However, although this new network exhibits similar global properties as the previous one, the lack of stability results in different genes being selected.” is unclear, consider rephrasing.

Discussion

In the sentence “Most of the network methods produced a relevant subset of biomarkers”, specify which methods.

In the sentence “Despite these similarities, the solutions were notably different.”, do you refer to the solutions found by all the methods?

In the sentence “While this deepens our understanding of the disease and provide biological hypotheses, they require further analyses”, give some examples of such analyses.

The sentence “Crucially, the consensus was as unstable as the tested methods, while the stable consensus shared these properties while accounting for instability.” is unclear, consider rephrasing.

“Another source of instability might be the redundancy inherent to biological networks” => why ?

“In section 2.3, we highlight ambiguities …”, I’m not sure 2.3 is the correct section and the following sentences are unclear. The “presented case” => which one ? “more resilient to them” => who ?

Comparison with the same approach on a larger network containing high-throughput and literature curated interactions. If the results are the same, how can they support the use of only interaction from HT experiments”? If the circular reasoning would be a problem, then the results would be different, no?

In the sentence “NEK10 (P-value 1.6 × 10 -5 , located near SLC4A7 ) or POU5F1B, were absent from the network”, are you talking about the Protein-Protein Interaction Network (PPIN) or the results you obtained? If it is about the PPIN, why didn’t you use a different one?

Specify phenotype in the sentence “However, on average protein-coding genes absent from the PPIN are less associated with this phenotype”.

In the sentence “As we are using interactions from high-throughput experiments, such difference…” it is unclear what the word “difference” refers to.

The sentence “On the other hand, Huang et al. [27] found that the best predictor of the performance of a network for disease gene discovery is the size of the network, which supports using the largest amount of interactions.” is unclear, consider rephrasing.

Add the name of such algorithms to the sentence “... P-value precision limited to the number of permutations [43]), and other algorithms [43, 29, 58]...”.

Materials and methods

The method section is clearer than the result section. Pay attention that some details are not given in the method section but elsewhere in the manuscript (for instance the used p-value thresholds...), and that not all the method sections are quoted in the results when necessary.

Section 4.1

Syntax: “We focused on the 2 577 samples of European ancestry, out of which 1 279 are controls and 1 298 are cases.”.

Section 4.3

4.3.1

The sentence “For the former, we considered any gene that can be mapped to any of the selected SNPs as selected as well.” is unclear, consider rephrasing.

4.3.3

“And they were designed to run efficiently on networks…” “they” is used to refer to 2 different things in the same sentence

4.3.4.

The sentence “The authors propose a transformation of the genes’ P-value into a score that is negative under no association with the phenotype, and positive when there is.” seems to be incomplete.

Add the meaning of “equilibrium” to the sentence “This process continues until equilibrium is reached...”.

Section 4.6.

The code lacks comments and it is difficult to reproduce the results from the paper.

Check broken links (I found this one https://github.com/hclimente/gwas-tools/blob/master)

Supplementary materials

Fig. 2.

The x-axis labels from 18 to 23 are broken in the three subfigures.

Fig. 4.

OFD1 and COPS5 are not highlighted in (B)

Fig. 5.

The gene names (Y-axis) are hard to read, increasing space between names.

Fig. 7.

Define all abbreviations.

Are there more than 5K protein coding genes missing in your PPIN? Did you consider using another one?

Fig. 8.

Add legend in a separate box to improve readability.

Reviewer #3: Network-based analysis of GWAS results is a potentially powerful approach for placing individual SNP/gene associations in the scaffold of molecular relationships to both improve interpretability and to the ability to identify marginally significant genes connected with significant genes. In this study, the authors perform an extensive analysis applying five methods for network-based GWAS analysis to study the associations from the GENESIS breast cancer study. The manuscript is written very well for the most part. The Introduction and Methods are particularly excellent as is the authors’ effort to release their analysis code. This is a valuable study but there are several concerns that need to be addressed.

1. The authors bring up several times the point that network-based methods tradeoff statistical significance for biological relevance.

- First, the logistic regression (LR) models based on the genes/SNPs from the network-based methods are not that predictive. Second, the original LR model they build using all SNPs also has SNPs with high p-values, which is cited as a motivation for using network-based methods. So, neither the motivation nor the benefit of this “tradeoff” is apparent.

- Similarly, why is solution subnetworks having lower p-values than the whole network (despite containing genes with higher p-values) exemplifying this tradeoff. The whole network contains the subnetwork and subnetworks of similar sizes can easily have lowe p-values by random chance.

2. In Table 1, please include the number of genes from each method that is included in the consensus network.

3. Association with network centrality:

- Genes in the four solution subnetworks exhibiting high between centrality: The dependence with betweenness shown in Table 1 seems to be a function of the number of genes in the subnetwork.

- Plus, in general, the association with high centrality genes is a chicken-and-egg problem.

- The same goes for the observations about the relationship with degree centrality (Figure 4B).

- The only way to check all these associations is to run all these analyses on a randomized PPI (where the edges have been shuffled while preserving node degrees) and compare to the real associations.

4. Figure 1:

- Figure 1C Precision-Recall analysis (comparison to BCAC): Is this analysis restricted to the set of common SNPs/genes? What is the interpretation? Both recall and precision here seem to be very low for all of the methods. The axes are a little confusing and also should be switched. Further, this is a chart talking about how BCAC genes relate to the methods. BCAC was never mentioned or defined in these early sections, so this does not make sense until one looks at this after reading much more of the paper. Same goes for BCAC mentioned in Figure 2.

- Comparing the network results to BCAC is a very useful validation metric. It would also be useful to run the network methods on the BCAC results themselves and compare the resulting networks from the different methods.

- Finally, it would be worth separating out figures relating to the BCAC comparison to the network methods from previous figures and putting them into this section. It would be more organized, make more sense rather than randomly seeing BCAC comparison in figure 1 where we are just being introduced to the network methods themselves.

5. Figure 3:

- 3A: Are the lines standard error in each direction? How can there be standard error in the direction of active set size since this seems to be fixed? What is the point of the grey dotted line?

- 3B: Patient classification using selected SNPs. This analysis is not helpful. The authors acknowledge that they do not expect to separate cases from controls just using genetic data. And, as expected the sensitivity and specificity values are all pretty close to 0.5 and very close to each other. So, this analysis is not useful.

- 3C: Does not have LEAN. Isn’t it intuitive that the more genes a network selects, the more stable that network is likely to be? Saying LEAN is stable when it never gets any significant genes seems to be a stretch. It is consistent in that it never gives results.

- 3D: Analysis of runtime is a little interesting but not useful at all. Runtime matters in the case of applying methods to a large number of traits/diseases. Since the application is to a single disease of interest, it should be reasonable if a method run for a day if it also returns reasonable/useful results. Also, if the additional 119,980 sec for VEGAS2 is a constant, then why does adding it to the method runtime result in different distances between the corresponding triangles and circles in Figure 3D.

6. Figure 4:

- Why is 4A labeled as a drawback?

- Doesn’t it make sense in 4B that most genes would have a low centrality? Most genes are not central hubs in networks. Is it necessarily a bad thing that there are significant genes that do not have a high centrality degree?

- 4C seems very oddly scaled. One assumes that each of these is a chromosome, and it makes it look like the genomic regions from each chromosome are the same size. It also seems very unclear what the background color means here. Also for example in chromosome 5 here, how many genes are being looked at? It looks like there are at least 3 genes here, but that is not clear at all. This also does not seem to coincide with the results from the SConES network described in previous sections

7. Figure 5:

- In Figure 5B, add another boxplot for genes picked by just one method.

- Figure 5C is not cited in the text.

8. Pathway enrichment analysis: Is the universe of genes should also be intersected with genes with at least one annotation in Reactome.

9. Figure S5C is very misleading. There are only 4 familial BC genes. This is not a very good image in general and does not tell us anything. Is there a comment on S5D where the familial genes don’t really have much of a trend?

**Have all data underlying the figures and results presented in the manuscript been provided?**

Reviewer #1: Yes

Reviewer #2: **No: **The GENESIS GWAS dataset used in the study is not available to the community

Reviewer #3: **No: **The data is not openly available.

PLOS authors have the option to publish the peer review history of their article (what does this mean?). If published, this will include your full peer review and any attached files.

Reviewer #1: No

Reviewer #2: No

Reviewer #3: No
---

## [Decision Letter · Decision Letter 1]

4 Jan 2021

Dear Dr. Climente-González,

Thank you very much for submitting your manuscript "Biological networks and GWAS: comparing and combining network methods to understand the genetics of familial breast cancer susceptibility in the GENESIS study" for consideration at PLOS Computational Biology. As with all papers reviewed by the journal, your manuscript was reviewed by members of the editorial board and by several independent reviewers. The reviewers appreciated the attention to an important topic. Based on the reviews, we are likely to accept this manuscript for publication, providing that you modify the manuscript according to the review recommendations.

Although the full GENESIS data might not be available, if summary statistics or GWAS associations can be released it would significantly benefit the readers and other researchers interested in this type of work.

Sincerely,

Sushmita Roy, Ph.D.

Deputy Editor

PLOS Computational Biology

Florian Markowetz

Deputy Editor

PLOS Computational Biology

[LINK]

Reviewer's Responses to Questions

**Comments to the Authors:**

Reviewer #2: The authors provide detailed responses to our previous comments as well as a revised version of their manuscript highly improved. The results are now much more clear and easy to follow, the interest of the work more straightforward, and figures have been improved.

Minor comments

Abstract and title are the two remaining points where there is still room for improvement in clarity. The title is a bit long, and the abstract has some of the previous weaknesses. In particular "project-agnostic" is unclear.

There seems to be problems in the figure numbering, as in the figure pages, the figure number title does not match with the figure number in the link, starting at figure 2, which is linked as figure3.

Introduction

"and aggregate them ..." is this really an aggregation? Aggregation could mean the union of the solution, whereas the consensus is using the intersection of the solutions.

Methods

The methods section describes 2 network topology measures, but only one median (betweenness) is presented in Table2 and discussed in the corresponding section of the results. Similarly, Figure 4 presents only the centrality measure but not the betweenness.

Network rewiring

“We only applied only four methods”, the word “only” is repeated.

2.4

"gene solutions" is unclear

3.2

A remaining unclear point is if the gene functional enrichment analysis is done individually on each subnetwork solution, or on the union of the genes from the different solutions.

"submodule" is unclear, is it a subnetwork solution?

3.6

Last part of the section, "While the solutions on those ... (Fig4C)", isn't it Figure 4D?

Some parts of the section “Network topology and association scores matter and might lead to ambiguous results” seem more appropriate for the discussion.

Figures

Figure 1

C. The networks corresponding to HotNet2 and SconES GI are overlapping with the legends.

D. Y-axis labels are missing

Figure 2

A. Y-axis labels are missing and Fig2A it is not referenced in the text

Figure 2B and 2C could be merged, the same information can be provided in the same network representation. Same observation for Figure 5B and 5C.

Figure 3

D. The axis’ legends (true positive and false positive rates) and the figure legend (true positive and true negative rates) do not match.

Figure 4

B. The legend of USHBP1 overlaps another one.

D. Not referenced in the text

Figure 5

A. Y-axis labels are missing

C. There is a line between the pink DNAJC1 legend and its corresponding node, which is a bit misleading since it is the only occurrence.

Figure S2.

C. Why none of the dots above the red line are labeled?

Figure S6. LEAN seems to be labeled as SconES GI or vice versa, plus it is referenced before figure S5.

Figure S7.

B. Seems to be truncated on the right

Figure S10. LC is not defined, plus it is referenced before figure S9.

Figure S11. Missing (the download link shows figure S10)

Discussion

“To overcome these limitations while exploiting the each method’s strengths…” syntax error, remove “the”

Acknowledgments

"clinics" duplicated

The text is not justified (alignment)

Reviewer #3: The authors have satisfactorily addressed all the questions and concerns raised by the reviewer.

**Have all data underlying the figures and results presented in the manuscript been provided?**

Reviewer #2: **No: **genetic data not available

Reviewer #3: **No: **The individual-level genotype-phenotype data is not openly available.

PLOS authors have the option to publish the peer review history of their article (what does this mean?). If published, this will include your full peer review and any attached files.

Reviewer #2: No

Reviewer #3: No
---

## [Decision Letter · Decision Letter 2]

18 Feb 2021

Dear Dr. Climente-González,

We are pleased to inform you that your manuscript 'Boosting GWAS using biological networks: a study on susceptibility to familial breast cancer' has been provisionally accepted for publication in PLOS Computational Biology.

Best regards,

Sushmita Roy, Ph.D.

Deputy Editor

PLOS Computational Biology

Florian Markowetz

Deputy Editor

PLOS Computational Biology

Reviewer's Responses to Questions

**Comments to the Authors:**

Reviewer #2: The authors addressed all our comments.

**Have all data underlying the figures and results presented in the manuscript been provided?**

Reviewer #2: Yes

PLOS authors have the option to publish the peer review history of their article (what does this mean?). If published, this will include your full peer review and any attached files.

Reviewer #2: No

---

## [Editor Report · Acceptance letter]

14 Mar 2021

PCOMPBIOL-D-20-00955R2 

Boosting GWAS using biological networks: a study on susceptibility to familial breast cancer

Dear Dr Climente-González,

I am pleased to inform you that your manuscript has been formally accepted for publication in PLOS Computational Biology. Your manuscript is now with our production department and you will be notified of the publication date in due course.

With kind regards,

Alice Ellingham
